# An approach to the verification of high-resolution ocean models using spatial methods

Ric Crocker[1], Jan Maksymczuk[2], Marion Mittermaier[1], Marina Tonani[2], Christine Pequignet[2]
[1]Verification, Impacts and Post-Processing, Weather Science, Met Office, Exeter, EX1 3PB, United Kingdom
[2]Ocean Forecasting Research & Development, Weather Science, Met Office, Exeter, EX1 3PB, United Kingdom
*Corresponding author:* ric.crocker@metoffice.gov.uk

# Abstract

The Met Office currently runs two operational ocean forecasting configurations for the North West European Shelf, an eddy-permitting model with a resolution of 7 km (AMM7), and an eddy-resolving model at 1.5 km (AMM15).

Whilst qualitative assessments have demonstrated the benefits brought by the increased resolution of AMM15, particularly in the ability to resolve finer-scale features, it has been difficult to show this quantitatively, especially in forecast mode. Application of typical assessment metrics such as the root mean square error have been inconclusive, as the high-resolution model tends to be penalised more severely, referred to as the double-penalty effect. This effect occurs in point-to point comparisons whereby features correctly forecast but misplaced with respect to the observations are penalised twice; once for not occurring at the observed location, and secondly for occurring at the forecast location, where they have not been observed.

An exploratory assessment of sea surface temperature (SST) has been made at in-situ observation locations using a single-observation-neighbourhood-forecast (SO-NF) spatial verification method known as the High-Resolution Assessment (HiRA) framework. The primary focus of the assessment was to capture important aspects of methodology to consider when applying the HiRA framework. Forecast grid points within neighbourhoods centred on the observing location are considered as pseudo ensemble members, so that typical ensemble and probabilistic forecast verification metrics such as the Continuous Ranked Probability Score (CRPS) can be utilised. It is found that through the application of HiRA it is possible to identify improvements in the higher resolution model which were not apparent using typical grid scale assessments.

This work suggests that future comparative assessments of ocean models with different resolutions would benefit from using HiRA as part of the evaluation process, as it gives a more equitable and appropriate reflection of model performance at higher resolutions.

# Keywords

35    verification, ocean forecasts, SST, spatial methods, neighbourhood, validation, double-penalty

# 1. Introduction

When developing and improving forecast models an important aspect is to assess whether model changes have truly improved the forecast. Assessment can be a mixture of subjective approaches, such as visualising forecasts and assessing whether the broad structure of a field is appropriate, or objective methods, comparing the difference between the forecast and an observed or analysed value of 'truth' for the model domain.

Different types of intercomparison can be applied to identify different underlying behaviours:

- between different forecasting systems over an overlapping region to check for model consistency between the two;
- between two versions of the same model to test the value of model upgrades prior to operational implementation;
- parent-son intercomparison, evaluating the impact of downscaling or nesting of models;
- a forecast comparison against reanalysis of the same model, inferring the effect of resolution and forcing, especially in coastal areas.

There are a number of works which have used these types of assessment to delve into the characteristics of forecast models (e.g. Aznar et al., 2015, Mason et al., 2019, Juza et al., 2015) and produce coordinated validation approaches (Hernandez et al., 2015).

To aid the production of quality model assessment, services exist which regularly produce multi-model assessments to deliver to the ocean community (e.g. Lorente et al., 2019a)

One of the issues faced when assessing high-resolution models against lower resolution models over the same domain is that often the coarser model appears to perform at least equivalently or better when using typical verification metrics such as root-mean-squared-error (RMSE) or mean error, which is a measure of the bias.  Whereas a higher resolution model has the ability and requirement to forecast greater variation, detail and extremes, a coarser model cannot resolve the detail and will, by its nature, produce smoother features with less variation resulting in smaller errors. This can lead to the situation that despite the higher resolution model looking more realistic it may verify worse (e.g. Mass et al., 2002**,** Tonani et al., 2019).

This is particularly the case when assessing forecast models categorically. If the location of a
feature in the model is incorrect then two penalties will be accrued, one for not forecasting the
feature where it should have been and one for forecasting the same feature where it did not
occur (the double penalty effect, e.g. Rossa et al., 2008). This effect is more prevalent in higher-
resolution models due to their ability to, at least, partially resolve smaller-scale features of
interest. If the lower resolution model could not resolve the feature, and therefore did not
forecast it, that model would only be penalised once. Therefore, despite giving potentially better
guidance the higher resolution model will verify worse.
Yet, the underlying need to quantitatively show the value of high-resolution led to the
development of so-called "spatial" verification methods which aimed to account for the fact the
forecast produced realistic features that were not necessarily at the right place or at quite the
right time (e.g. Ebert, 2008 or Gilleland, 2009). These methods have been in routine use within
the atmospheric model community for a number of years with some long-term assessments and
model comparisons (e.g. Mittermaier *et al.* 2013 for precipitation).
Spatial methods allow forecast models to be assessed with respect to several different types of
focus. Initially these methods were classified into four groups. Some methods look at the ability
to forecast specific features (e.g. Davis et al., 2006), some look at how well the model performs
at different scales (scale-separation, e.g. Casati et al., 2004). Others look at field deformation
(how much a field would have to be transformed to match a 'truth' field (e.g. Keil and Craig,
2007). Finally, there is neighbourhood verification, many of which are equivalent to low band-
pass filters. In these methods forecasts are assessed at multiple spatial or temporal scales to see
how model skill changes as the scale is varied.
Dorninger et al. (2018) provides an updated classification of spatial methods, suggesting a fifth
class of methods, known as distance metrics, which sit between field deformation and feature-
based methods. These methods evaluate the distances between features, but instead of just
calculating the difference in object centroids (which is typical), the distances between all grid
point pairs are calculated, which makes distance metrics more similar to field deformation
approaches. Furthermore, there is no prior identification of features. This makes distance metrics
a distinct group that warrants being treated as such in terms of classification.  Not all methods
are easy to classify. An example of this is the Integrated Ice Edge Error (IIEE) developed for
assessing the sea ice extent (Goessling et al., 2016).
This paper exploits the use of one such spatial technique for the verification of sea surface
temperature (SST), in order to determine the levels of forecast accuracy and skill across a range
of model resolutions. The High-Resolution Assessment framework (Mittermaier, 2014,
Mittermaier and Csima, 2017) is applied to the Met Office Atlantic Margin Model running at 7 km
(O'Dea et al., 2012, O'Dea et al., 2017, King et al., 2018) (AMM7), and 1.5 km (Graham et al.,
2018, Tonani et al., 2019) (AMM15) resolutions for the European North West Shelf (NWS).  The
aim is to deliver an improved understanding beyond the use of basic biases and RMS errors for
assessing higher resolution ocean models, which would then better inform users on the quality
of regional forecast products. Atmospheric science has been using high-resolution convective-
scale models for over a decade, and so have experience in assessing forecast skill on these scales,
so it is appropriate to trial these methods on eddy-resolving ocean model data. As part of the
demonstration, the paper also looks at how the method should be applied to different ocean
areas, where variation at different scales occurs due to underlying driving processes.

The paper was influenced by discussions on how to quantify the added value from investments
in higher resolution modelling given the issues around the double-penalty effect discussed above,
which is currently an active area of research within the ocean community (Lorente et al., 2019b,
Hernández et al., 2018, Mourre et al., 2019).
Section 2 describes the model and observations used in this study along with the method applied.
Section 3 presents the results, and section 4 discusses the lessons learnt while using HiRA on
ocean forecasts and sets the path for future work by detailing the potential and limitations of the
method.

## 118   2. Data and Methods

## 2.1 Forecasts

The forecast data used in this study are from the two products available in the Copernicus Marine Environment Monitoring Service (CMEMS, see e.g. Le Traon et al., 2019, for a summary of the service) for the North West European Shelf area:

- NORTHWESTSHELF_ANALYSIS_FORECAST_PHYS_004_001_b (AMM7)
- NORTHWESTSHELF_ANALYSIS_FORECAST_PHY_004_013 (AMM15)

The major difference between these two products is the horizontal resolution, ~7 km for AMM7 and 1.5 km for AMM15. Both systems are based on a forecasting ocean assimilation model with tides. The ocean model is NEMO (Nucleus for European Modelling of the Ocean, Madec, 2016), using the 3DVar NEMOVAR system to assimilate observations (Mogensen et al., 2012). These are surface temperature in-situ and satellite measurements, vertical profiles of temperature and salinity, and along track satellite sea level anomaly data. The models are forced by lateral boundary conditions from the UK Met Office North Atlantic Ocean forecast model and by the CMEMS Baltic forecast product BALTICSEA_ANALYSIS_FORECAST_PHY_003_006. The atmospheric forcing is given by the operational European Centre for Medium-Range Weather Forecasts (ECMWF) Numerical Weather Prediction model for AMM15, and by the operational UK Met Office Global Atmospheric model for AMM7.

|  | Resolution | Atmospheric forcing | Geographical model domain | |
|---|---|---|---|---|
| **AMM7** | ~7 km | MetUM 10 km | 40°N - 65°N | 20°W -13°E |
| **AMM15** | ~1.5 km | ECMWF IFS ~14 km | ~45°N - 63°N | ~20°W - 13°E |

Table 1: Summary of the main differences between NORTHWESTSHELF_ANALYSIS_FORECAST_PHYS_004_001_b (AMM7) and NORTHWESTSHELF_ANALYSIS_FORECAST_PHYS_004_013 (AMM15)

The AMM15 and AMM7 systems run once a day and provide forecasts for temperature, salinity, horizontal currents, sea level, mixed layer depth, and bottom temperature. Hourly instantaneous values and daily 25-hour, de-tided, averages are provided for the full water column.

AMM7 has a regular latitude-longitude grid, whilst AMM15 is computed on a rotated grid and re-gridded to have both models delivered to the (CMEMS) data catalogue

(http://marine.copernicus.eu/services-portfolio/access-to-products/) on a regular grid. A fuller
description of the respective configurations of the two models can be found in Tonani et al.,

147     (2019).


For the purposes of this assessment the 5-day daily mean sea surface potential temperature (SST)
forecasts (with lead times of 12, 36, 60, 84, 108 hours) were utilised for the period from January
to September 2019. Forecasts were compared for the co-located areas of AMM7 and AMM15.
Figure 1 shows the AMM7 and AMM15 co-located domain along with the land-sea mask for each
of the models. AMM15 has a more detailed coastline and SST field than AMM7 due to its higher
resolution. When comparing two models with different resolutions it is important to know
whether increased detail actually translates into better forecast skill. Additionally, the differences
in coastline representation can have an impact on any HiRA results obtained, as will be discussed
in a later section.

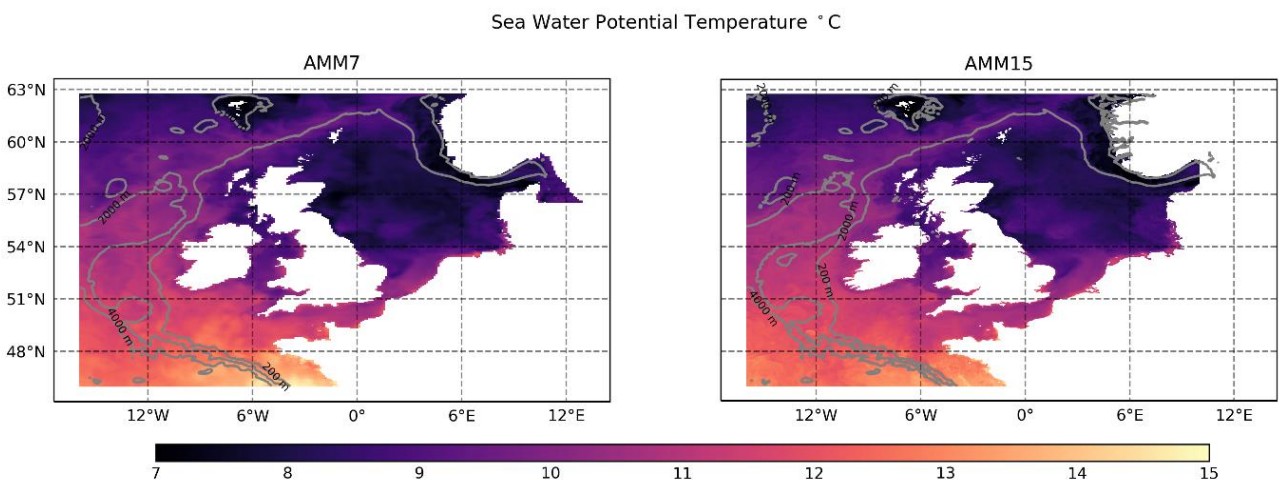


*Figure 1 - AMM7 and AMM15 co-located areas. Note the difference in the land-sea boundaries due to the different resolutions,*
*notably around the Scandinavian coast. Contours show the model bathymetry at 200, 2000 and 4000 m.*

It should be noted that this study is an assessment of the application of spatial methods to ocean
forecast data, and as such, is not meant as a full and formal assessment and evaluation of the
forecast skill of the AMM7 and AMM15 ocean configurations. To this end, a number of
considerations have had to be taken into account in order to reduce the complexity of this initial
study. Specifically, it was decided at an early stage to use daily mean SST temperatures, as
opposed to hourly instantaneous SST, as this avoided any influence of the diurnal cycle and tides
on any conclusions made. AMM15 and AMM7 daily means are calculated as means over 25 hours
to remove both the diurnal cycle and the tides. The tidal signal is removed because the period of
the major tidal constituent, the semidiurnal lunar component M2, is 12 hr and 25 min (Howarth
and Pugh, 1983). Daily means are also one of the variables that are available from the majority
of the products within the CMEMS catalogue, including reanalysis, so the application of the
spatial methods could be relevant in other use cases beyond those considered here. In addition,
there are differences in both the source and frequency of the air-sea interface forcing used in
both the AMM7 and AMM15 configurations which could influence the results. Most notably, the
AMM7 uses hourly surface pressure and 10 m winds from the Met Office Unified Model (UM),
whereas the AMM15 uses 3-hourly data from ECMWF.
## 2.2 Observations
SST observations used in the verification were downloaded from the CMEMS catalogue from the
product

- INSITU_NWS_NRT_OBSERVATIONS_013_036

This dataset consists of in-situ observations only, including daily drifters, mooring, ferry-box and
Conductivity Temperature Depth (CTD) observations. This results in a varying number of
observations being available throughout the verification period, with uneven spatial coverage
over the verification domain. Figure 2 shows a snapshot of the typical observational coverage, in
this case for 1200 UTC 6th June 2019. This coverage is important when assessing the results,
notably when thinking about the size and type of area over which an observation is meant to be
representative of, and how close to the coastline each observation is.

This study was set up to detect issues that should be considered by users when applying HiRA
within a routine ocean verification set-up, using a broad assessment containing as much data as
was available in order to understand the impact of using HiRA for ocean forecasts. Several
assumptions were made in this study.

For example, there is a temporal mismatch between the forecasts and observations used. The
forecasts (which were available at the time of this study) are daily means of the SSTs from 00 UTC
to 00 UTC, whilst the observations are instantaneous and usually available hourly. For the
purposes of this assessment, we have focused on SSTs closest to the mid-point of the forecast
period for each day (nominally 12 UTC). Observation times had to be within 90 minutes of this
time, with any other times from the same observation site being rejected. A particular reason for
picking a single observation time rather than daily averages was so that moving observations,
such as drifting buoys, could be incorporated into the assessment. Creating daily mean
observations from moving observations would involve averaging reports from different forecast
grid- boxes, and hence contaminate the signal that HiRA is trying to evaluate.

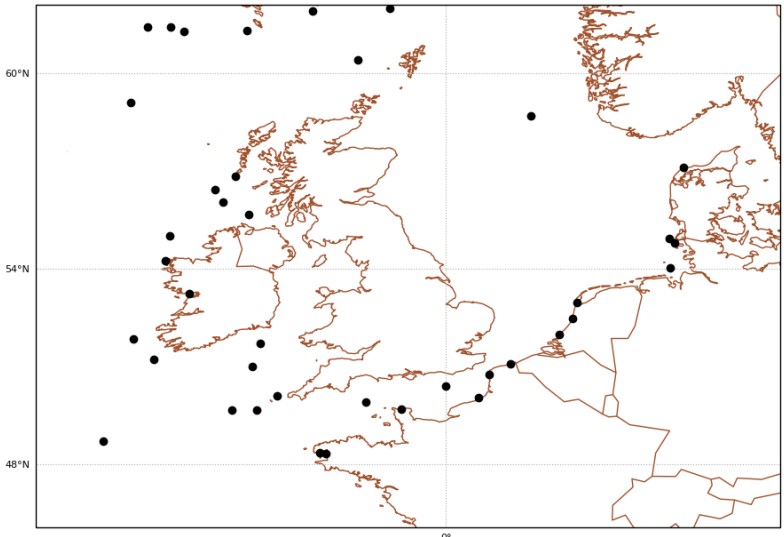


*Figure 2 - Observation locations within the domain for 1200 UTC on 6th June 2019.*
Future applications would probably contain a stricter set-up, e.g. only using fixed daily mean
observations, or verifying instantaneous (hourly) forecasts so as to provide a sub-daily
assessment of the variable in question.

3. High Resolution Assessment (HiRA)
The HiRA framework (Mittermaier, 2014) was designed to overcome the difficulties encountered
in assessing the skill of high-resolution models when evaluating against point observations.
Traditional verification metrics such as RMSE and mean error rely on a precise matching in space
and time, by (typically) extracting the nearest model grid point to an observing location. The
method is an example of a single-observation-neighbourhood-forecast (SO-NF) approach, with
no smoothing. All the forecast grid points within a neighbourhood centred on an observing
location are treated as a pseudo ensemble, which is evaluated using well known ensemble and
probabilistic forecast metrics. Scores are computed for a range of (increasing) neighbourhood
sizes to understand the scale-error relationship. This approach assumes that the observation is
representative of not only its precise location but also has characteristics of the surrounding area
as well. WMO manual No 8 (2017) suggests that, in the atmosphere, observations can be
considered to be representative of an area within a 100 km radius of a land station, but this is
often very optimistic. The manual states further: "For small-scale or local applications the
considered area may have dimensions of 10 km or less." A similar principle applies to the ocean,
i.e. observations can represent an area around the nominal observation location, though the
representative scales are likely to be very different from in the atmosphere. The representative
scale for an observation will also depend on local characteristics of the area, for example whether
the observation is on the shelf, or in open ocean or likely to be impacted by river discharge.
There will be a limit to the useful forecast neighbourhood size which can be used when comparing
to a point observation. This maximum neighbourhood size will depend on the representative
scale of the variable under consideration. Put differently, once the neighbourhoods become too
big there will be forecast values in the pseudo ensemble which will not be representative of the
observation (and the local climatology) and any skill calculated will be essentially random.
Combining results for multiple observations with very different representative scales (for
example a mixture of deep ocean and coastal observations) could contaminate results, due to
the forecast neighbourhood only being representative of a subset of the observations. The effect
of this is explored later in this paper.

HiRA can be based on a range of statistics, data thresholds and neighbourhood sizes in order to
assess a forecast model. When comparing deterministic models of different resolutions, the
approach is to equalise on the physical area of the neighbourhoods (i.e. having the same
"footprint"). By choosing sequences of neighbourhoods that provide (at least) approximate
equivalent neighbourhoods (in terms of area), two or more models can be fairly compared.
HiRA works as follows. For each observation, several neighbourhood sizes are constructed,
representing the length in forecast grid points of a square domain around the observation points,
centred on the grid point closest to the observation (Fig. 3). There is no interpolation applied to
the forecast data to bring it to the observation point, all the data values are used unaltered.




*Figure 3 - Example of forecast grid point selections for different HiRA neighbourhoods for a single observation point. A 3x3 domain*
*returns 9 points that represent the nearest forecast grid points in a square around the observation. A 5x5 domain encompasses*
*more points.*

Once neighbourhoods have been constructed, the data can be assessed using a range of well-
known ensemble or probabilistic scores. The choice of statistic usually depends on the
characteristics of the parameter being assessed. Parameters with significant thresholds can be
assessed using the Brier score (Brier, 1950) or the Ranked Probability Score (RPS) (Epstein, 1969),
i.e. assessing the ability of the forecast to correctly locate a forecast in the correct threshold
band. For continuous variables such as SST, the data has been assessed using the continuous
ranked probability score (CRPS) (Brown, 1974, Hersbach, 2000).
The CRPS is a continuous extension of the RPS. Whereas the RPS is effectively an average of a
user-defined set of Brier scores over a finite number of thresholds, the CRPS extends this by
considering an integral over all possible thresholds. It lends itself well to ensemble forecasts of
continuous variables such as temperature and has the useful property that the score reduces to
the mean absolute error (MAE) for a single grid point deterministic model comparison. This
means that if required, both deterministic and probabilistic forecasts can be compared using the
same score.
$$CRPS = \int_{-\infty}^{\infty} \left[ P_{fcst}(x) - P_{obs}(x) \right]^2 dx \quad (1)$$

Equation (1) defines the CRPS, where for a parameter x, $P_{fcst}(x)$ is the cumulative distribution of
the neighbourhood forecast and $P_{obs}(x)$ is the cumulative distribution of the observed value,
represented by a Heaviside function (see Hersbach, 2000). The CRPS is an error-based score
where a perfect forecast has a value of zero. It measures the difference between two cumulative
distributions, a forecast distribution formed by ranking the (in this case quasi) -ensemble
members represented by the forecast values in the neighbourhood, and a step function
describing the observed state. To use an ensemble, HiRA makes the assumption that all grid
points within a neighbourhood are equi-probable outcomes at the observing location. Therefore,
aside from the observation representativeness limit, as the neighbourhood sizes increase, this
assumption of equi-probability will break down as well, and scores become random. Care must
therefore be taken to decide whether a particular neighbourhood size is appropriately
representative. This decision will be based on the length scales appropriate for a variable as well
as the resolution of the forecast model being assessed. Figure 4 shows a schematic of how
different neighbourhood sizes contribute towards constructing forecast probability density
functions around a single observation.

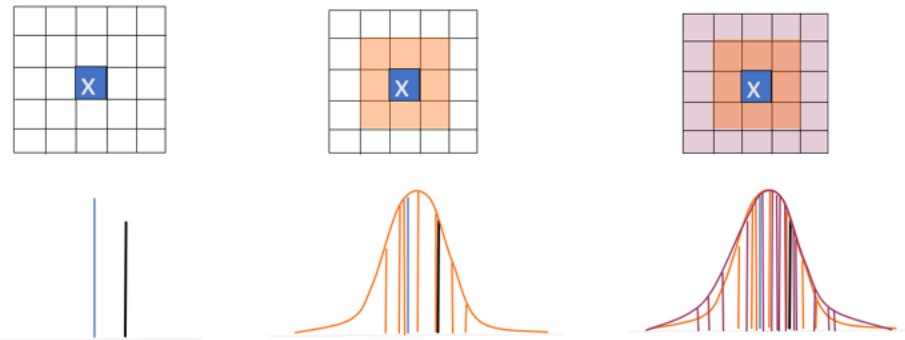

*Figure 4 – Example of how different forecast neighbourhood sizes would contribute to generation of a probability density function*
*around an observation (denoted by x). The larger the neighbourhood, the better described the pdf, though potentially at the*
*expense of larger spread. Where a forecast point is invalid within the forecast neighbourhood then that site is rejected from the*
*calculations for that neighbourhood size.*

AMM7 and AMM15 resolve different length scale of motion, due to their horizontal resolution.
This should be taken into account when assessing the results of different neighbourhood sizes.
Both models can resolve the large barotropic scale (~200 km) and the shorter baroclinic scale off
the shelf, in deep water. On the continental shelf, only the resolution of ~1.5 km of AMM15,
permits motions at the smallest baroclinic scale since the first baroclinic Rossby radius is of order
of 4 km (O'Dea et al., 2012). AMM15 represents a step change in representing the eddy dynamics
variability on the continental shelf. This difference has an impact also on the data assimilation
scheme, where two horizontal correlation length scales (Mirouze et al., 2016) are used to
represent large and small scales of ocean variability. The long length scale is 100 km while the
short correlation length scale aims to account for internal ocean processes variability,
characterized by the Rossby radius of deformation. Computational requirements restrict the
short length scale to be at least 3 model grid points, 4.5 km and 21 km respectively for AMM15
and AMM7 (Tonani et al., 2019). Although AMM15 resolves smaller scale processes, comparing
AMM7 and AMM15 in neighbourhood sizes between the AMM7 resolution and multiples of this
resolution will address processes that should be accounted for in both models.

As the methodology is based on ensemble and probabilistic metrics it is naturally extensible to
ensemble forecasts (see Mittermaier and Csima, 2017), which are currently being developed in
research-mode by the ocean community, allowing for inter-comparison between deterministic
and probabilistic forecast models in an equitable and consistent way.

## 4. Model Evaluation Tools (MET)
Verification was performed using the Point-Stat tool, which is part of the Model Evaluation Tools
(MET) verification package, that was developed by the National Center for Atmospheric Research
(NCAR), and which can be configured to generate CRPS results using the HiRA framework. MET is
free to download from GitHub at https://github.com/NCAR/MET.

## 5. Equivalent neighbourhoods and equalisation
When comparing neighbourhoods between models, the preference is to look for similar–sized
areas around an observation and then transforming this to the closest odd-numbered, square
neighbourhood, which will be called the 'equivalent neighbourhood'. In the case of the two
models used, the most appropriate neighbourhood size can change depending on the structure
of the grid so the user needs to take into consideration what is an accurate match between the
models being compared.

The two model configurations used in this assessment are provided on standard latitude-
longitude grids via the CMEMS catalogue. The AMM7 and AMM15 configurations are stated to
have resolutions approximating 7 km and 1.5 km respectively. Thus, equivalent neighbourhoods
should simply be a case of matching neighbourhoods with similar spatial distances. In fact, the
AMM15 is originally run on a rotated latitude-longitude grid where the resolution is closely
approximated by 1.5 km and subsequently provided to the CMEMS catalogue on the standard
latitude-longitude grid. Once the grid has been transformed to a regular latitude-longitude grid
the 1.5 km nominal spatial resolution is not as accurate. This is particularly important when
neighbourhood sizes become larger, since any error in the approximation of the resolution will
become multiplied as the number of points being used increases.

Additionally, the two model configurations do not have the same aspect ratio of grid points.
AMM7 has a longitudinal resolution of ~0.11° and a latitudinal resolution of ~0.066° (a ratio of
3:5) whilst the AMM15 grid has a resolution of ~0.03° and ~0.0135° respectively (a ratio of 5:11).
HiRA neighbourhoods typically contain the same number of grid-points in the zonal and
meridional directions which will lead to discrepancies in the area selected when comparing
models with different grid aspect ratios, depending on whether the comparison is based on
neighbourhoods with a similar longitudinal or similar latitudinal size. This difference will scale as
the neighbourhood size increases as shown in Fig. 4 and Table 2. The onus is therefore on the
user to understand any difference in grid structure, and therefore within the HiRA
neighbourhoods, between models being compared and to allow for this when comparing
equivalent neighbourhoods.


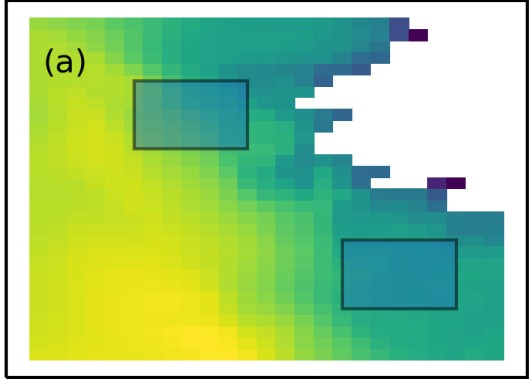 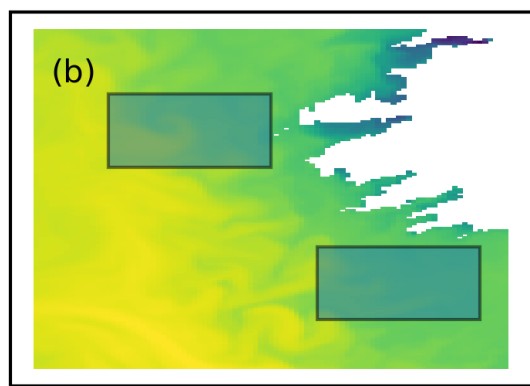


*Figure 5 - Similar neighbourhood sizes for a 49 km neighbourhood using the approximate resolutions (7 km and 1.5 km) with a) AMM7 with a 7x7 neighbourhood, b) AMM15 with a 33x33 neighbourhood. Whilst the neighbourhoods are similar sizes in the latitudinal direction, the AMM15 neighbourhood is sampling a much larger area due to different scales in the longitudinal direction. This means that a comparison with a 25x25 AMM15 neighbourhood is more appropriate.*

*Table 2 - Details of equivalent neighbourhoods used when comparing AMM7 and AMM15.*

| Name | AMM7 | | | | AMM15 | | | |
| | Total Points | Shape | Size (E-W) | | Total Points | Shape | Size (E-W) | |
| | | | *Actual (°)* | *Nominal (km)* | | | *Actual (°)* | *Nominal (km)* |
|---|---|---|---|---|---|---|---|---|
| **NB1** | 1 | 1x1 | 0.11 | 7 | 25 | 5x5 | 0.15 | 7.5 |
| **NB2** | 9 | 3x3 | 0.33 | 21 | 121 | 11x11 | 0.33 | 16.5 |
| **NB3** | 25 | 5x5 | 0.55 | 35 | 361 | 19x19 | 0.57 | 28.5 |
| **NB4** | 49 | 7x7 | 0.77 | 49 | 625 | 25x25 | 0.76 | 37.5 |
| **NB5** | 81 | 9x9 | 0.99 | 63 | 1089 | 33x33 | 0.99 | 49.5 |

359

For this study we have matched neighbourhoods between model configurations based on their longitudinal size. The equivalent neighbourhoods used to show similar areas within the two configurations are indicated in Table 2 along with the bar style and naming convention used throughout.


For ocean applications there are other aspects of the processing to be aware of when using neighbourhood methods. This is mainly related to the presence of coastlines and how their representation changes resolution (as defined by the land-sea mask) and the treatment of

observations within HiRA neighbourhoods. Figure 5 illustrates the contrasting land-sea
boundaries due to the different resolutions of the two configurations. When calculating HiRA
neighbourhood values, all forecast values in the specific neighbourhood around an observation
must be present for a score to be calculated. If any forecast points within a neighbourhood
contain missing data then that observation at that neighbourhood size is rejected. This is to
ensure that the resolution of the "ensemble", which is defined or determined by the number of
members, remains the same. For typical atmospheric fields such as screen temperature this is
not an issue, but with parameters that have physical boundaries (coastlines), such as SST, there
will be discontinuities in the forecast field that depend on the location of the land-sea boundary.
For coastal observations, this means that as the neighbourhood size increases, it is more likely
that an observation will be rejected from the comparison due to missing data. Even at the grid
scale, the nearest model grid point to an observation may not be a sea point. In addition, different
land-sea borders between models mean that potentially some observations will be rejected from
one model comparison but will be retained in the other because of missing forecast points within
their respective neighbourhoods. Care should be taken when implementing HiRA to check the
observations available to each model configuration when assessing the results and make a
judgement as to whether the differences are important.
There are potential ways to ensure equalisation, for example only using observations that are
available in both configurations for a location and neighborhoods, or only observations away
from the coast. For the purposes of this study, which aims to show the utility of the method, it
was judged important to use as many observations as possible, so as to capture any potential
pitfalls in the application of the framework, which would be relevant to any future application of
it.

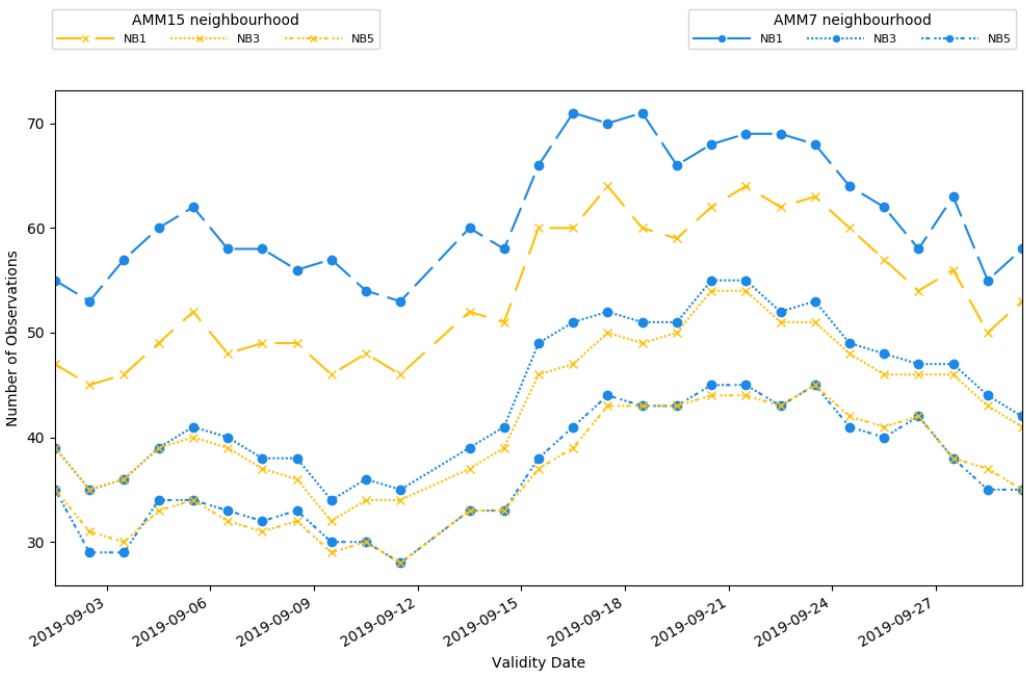

*Figure 6- Number of observation sites within NB1, NB3 and NB5 for AMM15 and AMM7. Numbers are those used during September 2019 but represent typical total observations during a month. Matching line styles represent equivalent neighbourhoods.*

Figure 6 shows the number of observations available to each neighbourhood for each day during September 2019. For each model configuration it shows how these observations vary within the HiRA framework. There are several reasons for the differences shown in the plot. There is the difference mentioned previously whereby a model neighbourhood includes a land point, and therefore is rejected from the calculations because the number of quasi-ensemble members is no longer the same. This is more likely for coastal observations and depends on the particularities of the model land-sea mask near each observation. This rejection is more likely for the high-resolution AMM15 when looking at equivalent areas, in part due to the larger number of grid boxes being used; however, there are also instances of observations being rejected from the coarser resolution AMM7 and not the higher-resolution AMM15 due to nuances of the land-sea mask.

It is apparent that for equivalent neighbourhoods there are typically more observations available
for the coarser model configuration and that this difference is largest for the smallest equivalent
neighbourhood size but becoming less obvious at larger neighbourhoods. It could therefore be
worth considering that the large benefit in AMM15 when looking at the first equivalent
neighbourhood is potentially influenced by the difference in observations. As the neighbourhood
sizes increase, the number of observations reduces due to the higher likelihood of a land point
being part of a larger neighbourhood. It is also noted that there is a general daily variability in the
number of observations present, based on differences in the observations reporting on any
particular day within the co-located domain.

## 6. Results

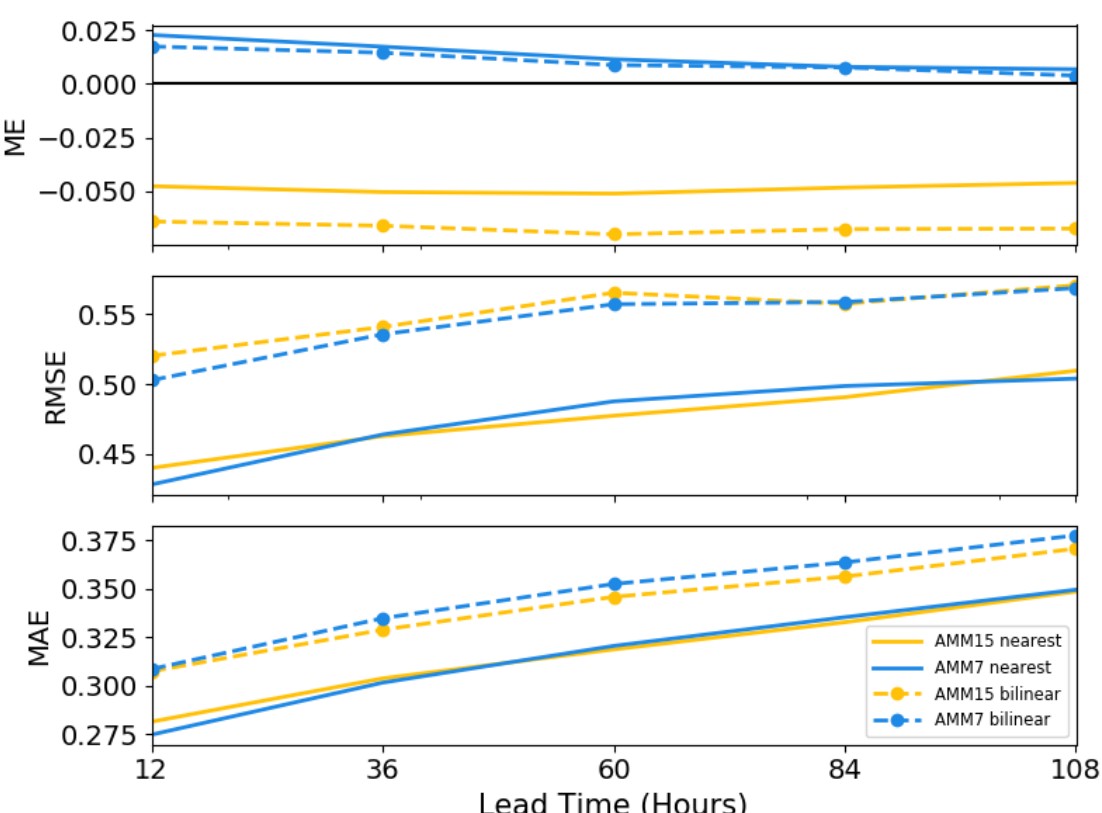


*Figure 7 - Verification results using a typical statistics approach for January – September 2019. Mean error (top), root mean square*
*error (middle) and mean absolute error (bottom) results are shown for the two model configurations. Two methods of matching*
*forecast to observations points have been used; a nearest neighbor approach (solid) representing the single grid point results from*
*HiRA, and a bilinear interpolation approach (dashed) more typically used in operational ocean verification.*
Figure 7 shows the aggregated results from the study period defined in Section 2 by applying
typical verification statistics. Results have been averaged across the entire period from January
to September and output relative to the forecast validity time. Two methods of matching forecast
grid points to observation locations have been used. Bilinear interpolation is typically the
approach used in traditional verification of SST, as it is a smoothly varying field. A nearest
neighbour approach has also been shown, as this is the method that would be used for HiRA
when applying it at the grid scale.
It is noted that the two methods of matching forecasts to observation locations give quite
different results. For the mean error, the impact of moving from a single grid point approach to
a bilinear interpolation method appears to be minor for the AMM7 model, but is more severe for
the AMM15, resulting in a larger error across all lead times. For the RMSE the picture is more
mixed, generally suggesting that the AMM7 forecasts are better when using a bilinear
interpolation method but giving no clear overall steer when the nearest grid point is used.
However, the impact of taking a bilinear approach results in much higher gross errors across all
lead times when compared to the nearest grid point approach.
The MAE has been suggested as a more appropriate metric than the RMSE for ocean fields using
(as is the case here) near real time observation data (Brassington, 2017). In Fig. 6 it can be seen
that the nearest grid point approach for both AMM7 and AMM15 gives almost exactly the same
results, except for the shortest of lead times. For the bilinear interpolation method, AMM15 has
a smaller error than AMM7 as lead time increases, behavior which is not apparent when RMSE is
applied.
Based on the interpolated RMSE results in Fig. 6 it would be hard to conclude that there was a
significant benefit to using high-resolution ocean models for forecasting SSTs. This is where the
HiRA framework can be applied. It can be used to provide more information, which can better
inform any conclusions on model error.


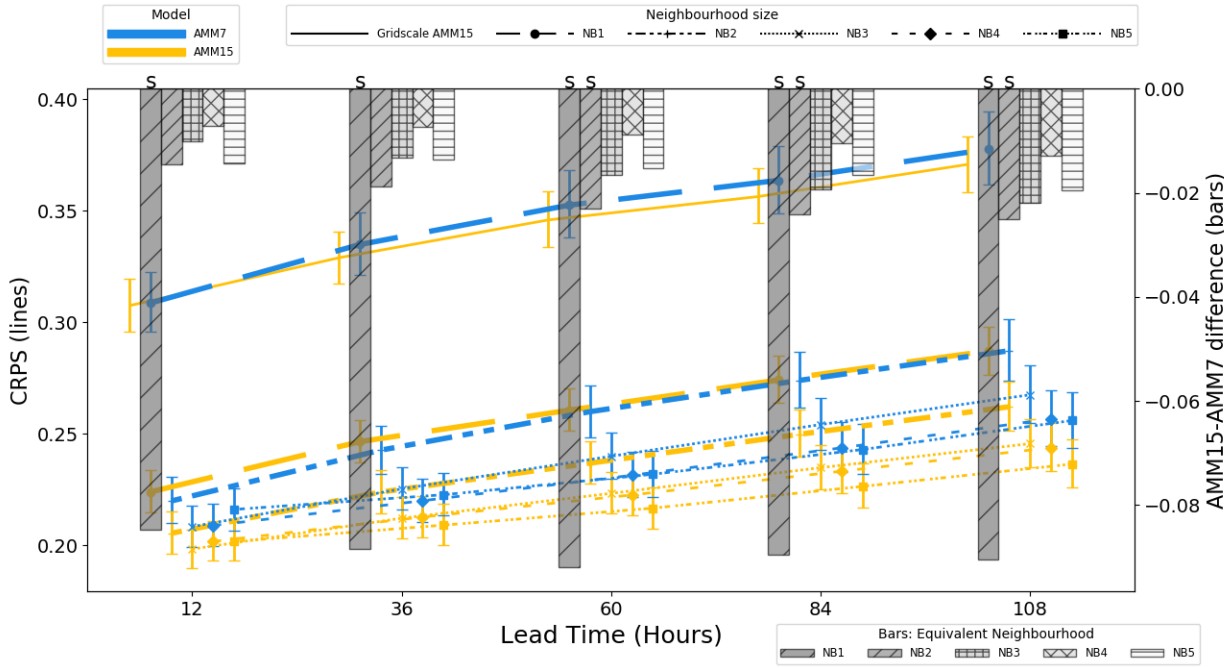


*Figure 8- Summary of CRPS (left axis, lines) and CRPS difference (right axis, bars) for the period January 2019 to September 2019*

*for AMM7 and AMM15 models at different neighbourhood sizes. Error bars represent 95 % confidence intervals generated using*

*a bootstrap with replacement method for 10000 samples. An 'S' above the bar denotes that 95 % error bars for the two models*

*do not overlap.*

Figure 8 shows the results for AMM7 and AMM15 for the period January - September 2019 using

the HiRA framework with the CRPS. The lines on the plot show the CRPS for the two model

configurations for different neighbourhood sizes, each plotted against lead-time. Similar line

styles are used to represent equivalent neighbourhood sizes. Confidence intervals have been

generated by applying a bootstrap with replacement method, using 10000 samples, to the

domain-averaged CRPS (e.g. Efron and Tibshirani, 1993). The error bars represent the 95 %

confidence level. The results for the single grid-point show the MAE and are the same as would

be obtained using a traditional (precise) matching. In the case of CRPS, where a lower score is

better, we see that AMM15 is better than AMM7, though not significantly so, except at shorter

lead-times where there is little difference.

The differences at equivalent neighbourhood sizes are displayed as a bar plot on the same figure,

with scores referenced with respect to the right-hand axis. Line markers and error bars have been

offset to aid visualization, such that results for equivalent neighbourhoods are displayed in the

same vertical column as the difference indicated by the barplot. The details of the equivalent
neighbourhood sizes are presented in Table 2. Since a lower CRPS score is better, a positively
orientated (upwards) bar implies AMM7 is better, whilst a negatively orientated (downwards)
bar means AMM15 is better.
As indicated in Table 2, NB1 compares the single grid-point results of AMM7 with a 25-member
pseudo-ensemble constructed from a 5x5 AMM15 neighbourhood. Given the different
resolutions of the two configurations, these two neighbourhoods represent similar physical areas
from each model domain, with AMM7 only represented by a single forecast value for each
observation, but AMM15 represented by 25 values cover the same area, and as such potentially
better able to represent small-scale variability within that area.
At this equivalent scale the AMM15 results are markedly better than AMM7, with lower errors,
suggesting that overall the AMM15 neighbourhood better represents the variation around the
observation than the coarser single grid point of AMM7. At the next set of equivalent
neighbourhoods (NB2), the gap between the two configurations has closed, but AMM15 is still
consistently better than AMM7 as lead time increases.  Above this scale the neighbourhood
values tend towards similarity, and then start to diverge again suggesting that the representative
scale of the neighbourhoods has been reached and that errors are essentially random.
Whilst the overall HiRA neighbourhood results for the co-located domains appear to show a
benefit to using a higher resolution model forecast, it could be that these results are influenced
by the spatial distribution of observations within the domain and the characteristics of the
forecasts at those locations. In order to investigate whether this was important behaviour, the
results were separated into two domains, one representing the continental shelf part of the
domain (where the bathymetry < 200 m), and the other representing the deeper, off-shelf, ocean
component (Fig. 8). HiRA results were compared for observations only within each masked
domain.

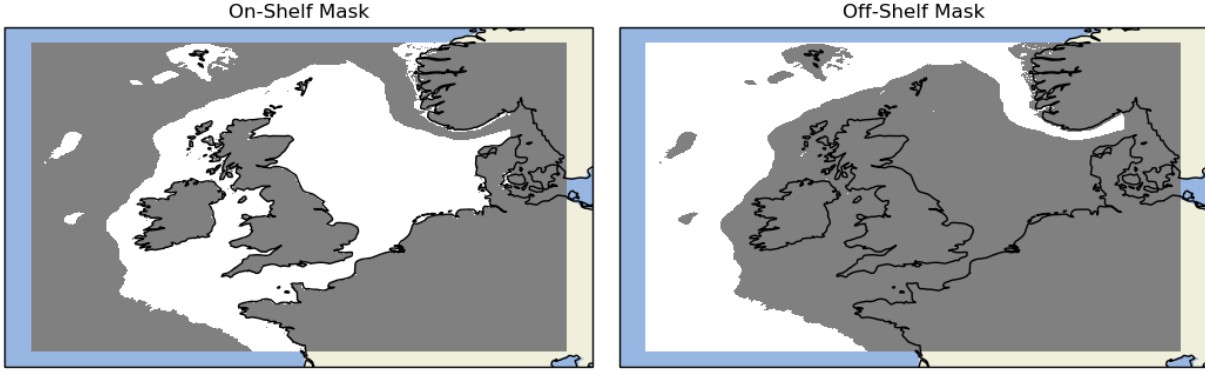


*Figure 9 - On-shelf and off-shelf masking regions within the co-located AMM7 and AMM15 domain (data within the grey areas is masked).*


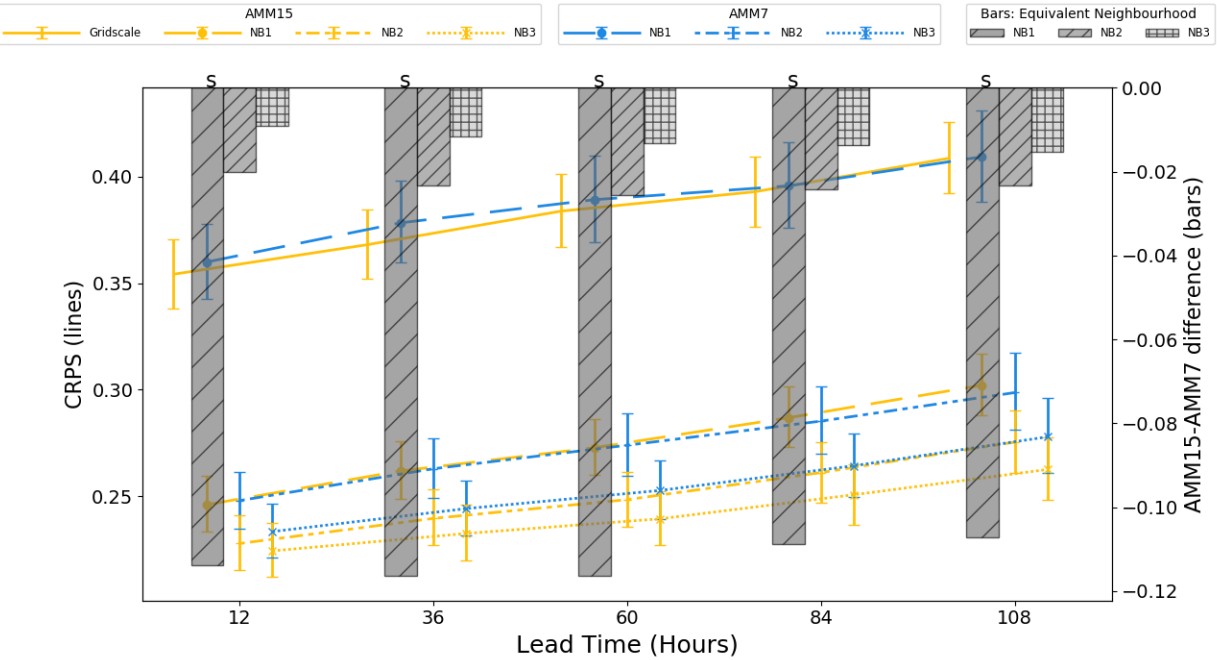


*Figure 10- Summary of on-shelf CRPS (left axis, lines) and CRPS difference (right axis, bars) for the period January 2019 to September 2019 for AMM7 and AMM15 models at different neighbourhood sizes. Error bars represent 95 % confidence values obtained from 10000 samples using bootstrap with replacement. An 'S' above the bar denotes that 95 % error bars for the two models do not overlap.*

On-shelf results (Fig. 10) show that at the grid scale the results for both AMM7 and AMM15 are
worse for this sub-domain. This could be explained by both the complexity of processes (tides,
friction, river mixing, topographical effects, etc.), and the small dynamical scales associated with
shallow waters on the shelf (Holt et al., 2017).

The on-shelf spatial variability in SST across a neighbourhood is likely to be higher than for an
equivalent deep ocean neighbourhood due to small-scale changes in bathymetry, and for some
observations, the impact of coastal effects. Both AMM7 and AMM15 show improvement in CRPS
with increased neighbourhood size until the CRPS plateaus in the range 0.225 to 0.25, with
AMM15 generally better than AMM7 for equivalent neighbourhood sizes. Scores get worse
(errors increase) for both model configurations as the forecast lead time increases.


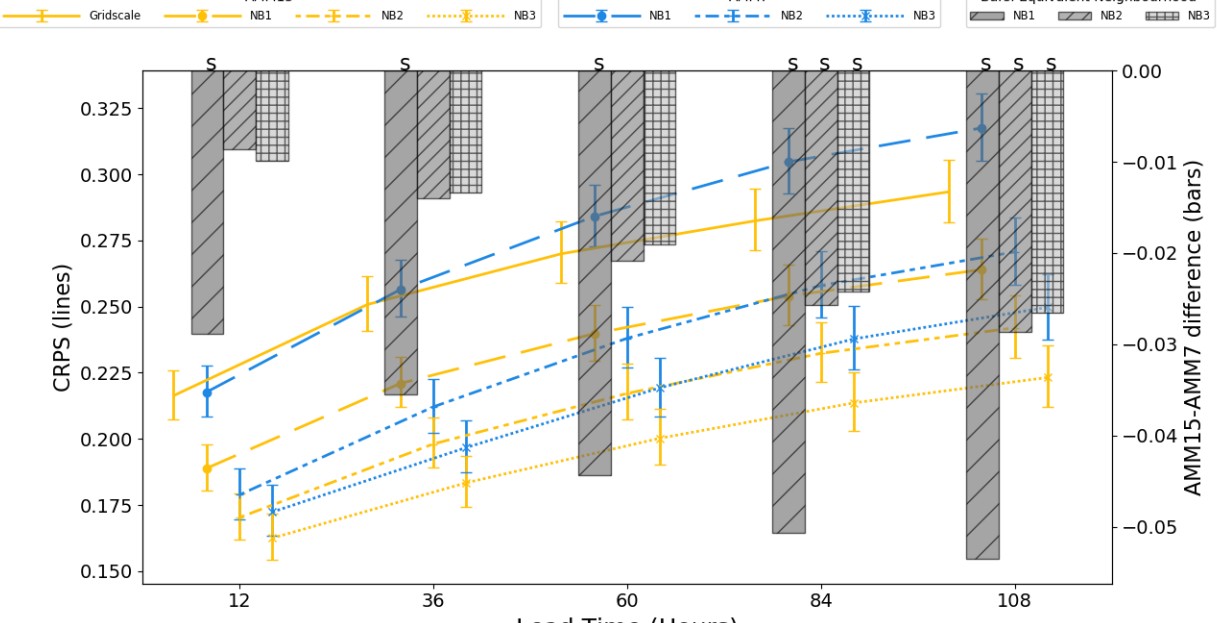

*Figure 11 – Summary of off-shelf CRPS (left axis, lines) and CRPS difference (right axis, bars) for the period January 2019 to*
*September 2019 for AMM7 and AMM15 models at different neighbourhood sizes. Error bars represent 95 % confidence values*
*obtained from 10000 samples using bootstrap with replacement. An 'S' above the bar denotes that 95 % error bars for the two*
*models do not overlap.*

For off-shelf results (Fig. 11), the CRPS is much better (smaller error), at both the grid scale and for HiRA neighbourhoods, suggesting that both configurations are better at forecasting these deep ocean SSTs (or that it is easier to do so). There is still an improvement in CRPS when going from the grid scale (single grid box) to neighbourhoods, but the value of that change is much smaller than for the on-shelf sub-domain. When comparing equivalent neighbourhoods, the AMM15 still gives consistently better results (smaller errors) and appears to improve over AMM7 as lead time increases in contrast to the on-shelf results.

It is likely that the neighbourhood at which we lose representativity will be larger for the deeper ocean than the shelf area because of the larger scale of dynamical processes in deep water. When choosing an optimum neighbourhood to use for assessment, care should be taken to check whether there are different representativity levels in the data (such as here for on-shelf and off-shelf) and pragmatically choose the smaller of those equivalent neighbourhoods when looking at data combining the different representativity levels.

Overall, for the period January-September 2019, the AMM15 demonstrates a lower (better) CRPS than AMM7 when looking at the HiRA neighbourhoods. However, this also appears to be true at the grid scale over the assessment period. One of the aspects that HiRA is trying to provide additional information about is whether higher resolution models can demonstrate improvement over coarser models against a perception that the coarser models score better in standard verification forecast assessments. Assessed over the whole period, this initial premise does not appear to hold true, therefore a deeper look at the data is required to assess whether this signal is consistent within shorter time periods, or whether there are underlying periods contributing significant and contrasting results to the whole-period aggregate.

Figure 12 shows a monthly breakdown of the grid scale and the NB2 HiRA neighbourhood scores at T+60. This shows the underlying monthly variability not immediately apparent in the whole-period plots. Notably for the January to March period, AMM7 outperforms AMM15 at the grid scale. With the introduction of HiRA neighbourhoods, AMM7 still performs better for February and March but the difference between the models is significantly reduced. For these monthly

timeseries the error bars increase in size relative to the summary plots (e.g. Fig 8) due to the
reduction in data available. The sample size will have an impact on the error bars as the smaller
the sample, the less representative of the true population the data is likely to be. April in
particular contained several days of missing forecast data, leading to a reduction in sample size
and corresponding increase in error bar size, whilst during May there was a period with reduced
numbers of observations.

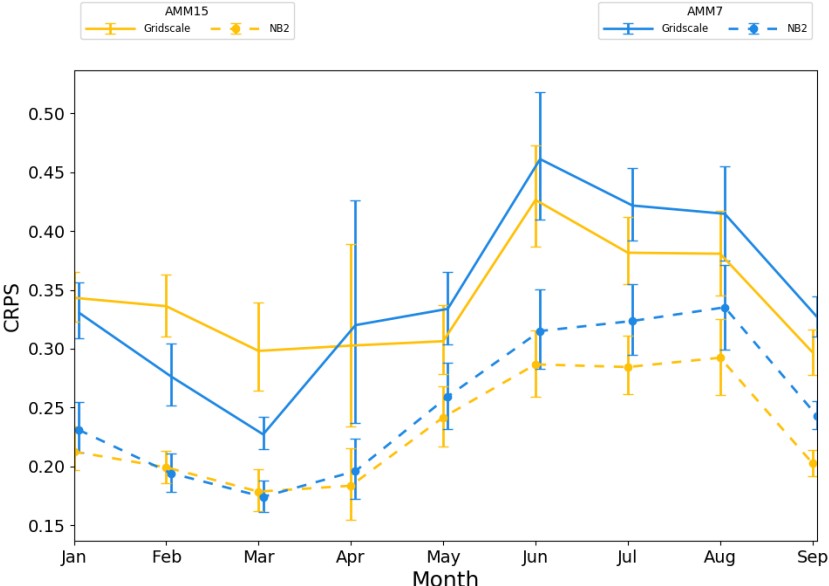


*Figure 12 – Monthly time series of whole-domain CRPS scores for grid scale (solid line) and NB2 neighbourhood (dashes) for T+60*
*forecasts. Error bars represent 95 % confidence values obtained from 10000 samples using bootstrap with replacement. Error bars*
*have been staggered in the x-direction to aid clarity.*

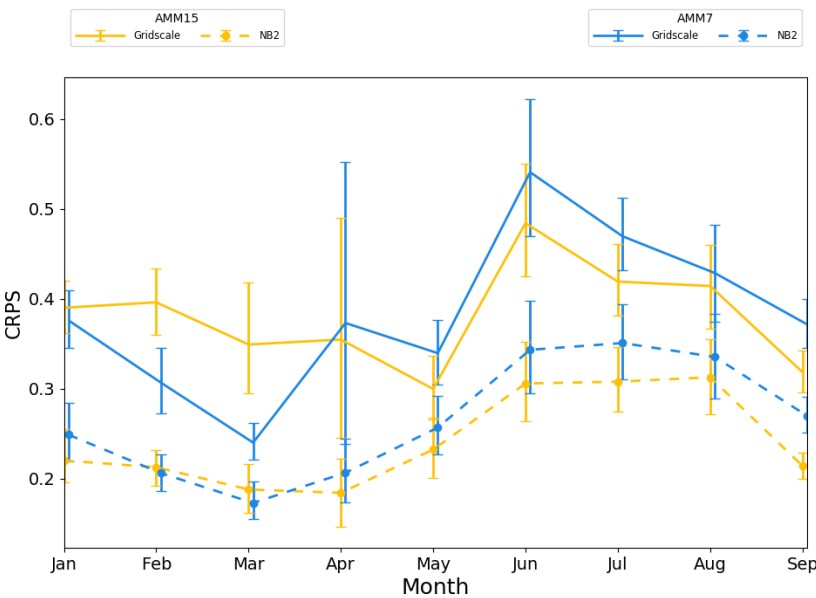


*Figure 13 - On-shelf monthly time series of CRPS. Error bars represent 95 % confidence values obtained from 10000 samples using bootstrap with replacement.*



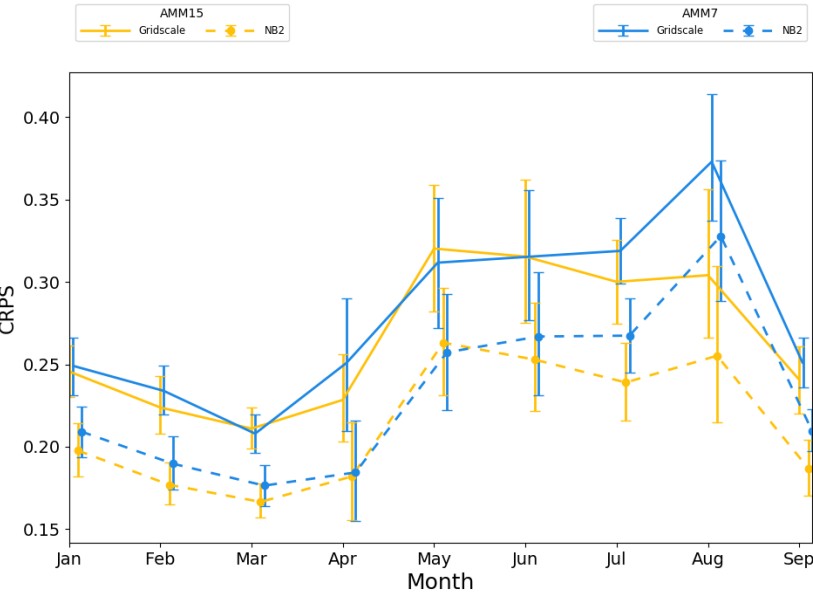


*Figure 14 - Off-shelf monthly time series of CRPS. Error bars represent 95 % confidence values obtained from 10000 samples using bootstrap with replacement.*



The same pattern is present for the on-shelf sub-domain (Fig. 13), where what appears to be a
significant benefit for the AMM7 during February and March is less clear-cut at the NB2
neighbourhood. For the off-shelf sub-domain (Fig. 14), differences between the two
configurations at the grid scale are mainly apparent during the summer months. At the NB2 scale,
the AMM15 potentially demonstrates more benefit than AMM7 except for April and May, where
the two show similar results.  There is a balance to be struck in this conclusion as the differences
between the two models are rarely greater than the 95 % error bars. This in itself does not mean
that the results are not significant. However, care should be taken when interpreting such a result
as a statistical conclusion rather than broad guidance as to model performance. Attempts to
reduce the error bar size, such as increasing the number of observations, or number of times
within the period would aid this interpretation.
One noticeable aspect of the time series plots is that the whole-domain plot is heavily influenced
by the on-shelf results. This is due to the difference in observation numbers as shown in Fig. 15,
with the on-shelf domain having more observations overall, sometimes significantly more, for
example during January or mid-late August. For the overall domain, the on-shelf observations
will contribute more to the overall score and hence the underlying off-shelf signal will tend to be
masked. This is an indication of why verification is more useful when done over smaller, more
homogeneous sub-regions, rather than verifying everything together, with the caveat that
sample sizes are large enough, since underlying signals can be swamped by dominant error types.

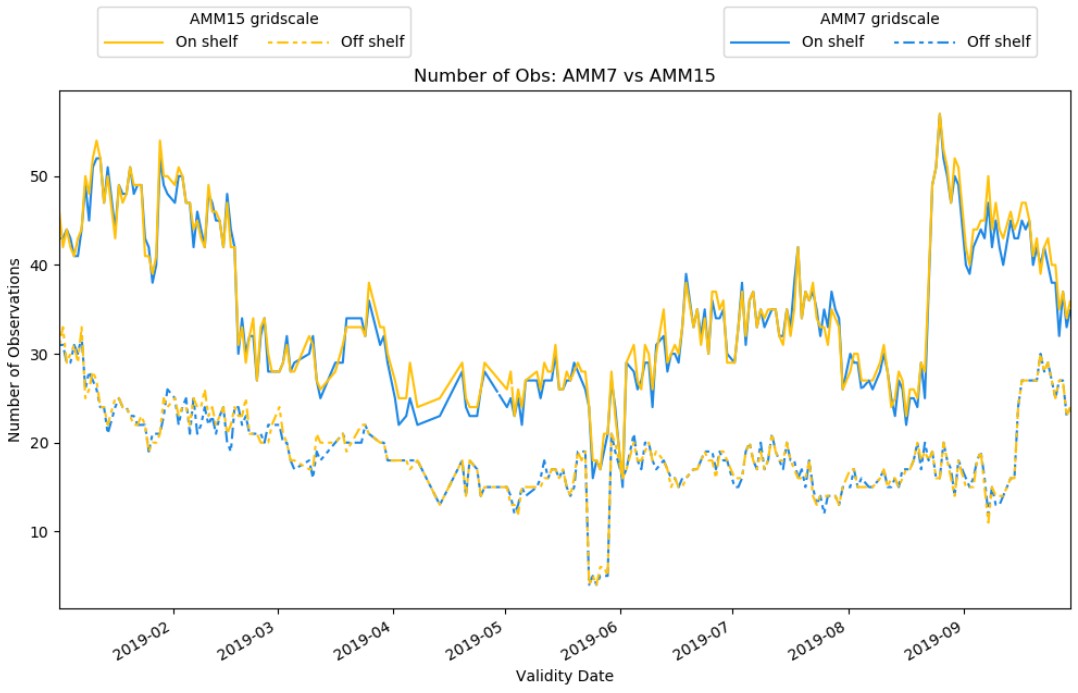


*Figure 15 - Number of grid scale observations for the on and off-shelf domains.*

## 7. Discussion and Conclusions

In this study, the HiRA framework has been applied to SST forecasts from two ocean models with
different resolutions. This enables a different view of the forecast errors than obtained using
traditional (precise) grid scale matching against ocean observations. Particularly it enables us to
demonstrate the additional value of high-resolution model. When considered more
appropriately high-resolution models (with the ability to forecast small-scale detail) have lower
errors when compared to the smoother forecasts provided by a coarser-resolution model.
The HiRA framework was intended to address the question 'Does moving to higher resolution
add value?' This study has identified and highlighted aspects that need to be considered when
setting up such an assessment. Prior to this study, routine verification statistics typically showed
that coarser resolution models had equivalent or more skill than higher resolution models (e.g.
Mass et al., 2002**,** Tonani et al., 2019).  During the period January to September 2019, grid scale
verification within this assessment showed that the coarser-resolution AMM7 often
demonstrated lower errors than the AMM15.
HiRA neighbourhoods were applied and the data then assessed using the CRPS, showing a large
reduction (improvement) in errors for AMM15 when going from a grid scale, point-based
verification assessment to a neighbourhood, ensemble approach. When applying an equivalent-
sized neighbourhood to both configurations, AMM15 typically demonstrated lower (better)
scores. These scores were in turn broken down into off-shelf and on-shelf sub-domains and
showed that the different physical processes in these areas affected the results. Forecast
verification studies tailored for the coastal/shelf areas are needed to properly understand the
forecast skills in areas with high complexity and fast evolving dynamics.
When constructing HiRA neighbourhoods the spatial scales that are appropriate for the
parameter must be considered carefully. This often means running at several neighbourhood
sizes and determining where the scores no longer seem physically representative. When
comparing models, care should be taken to construct neighbourhood sizes that are similarly sized
spatially, the details of the neighbourhood sizes will depend on the structure and resolution of
the model grid.
Treatment of observations is also important in any verification set-up. For this study, the fact that
there are different numbers of observations present at each neighbourhood scale (as
observations are rejected due to land contamination) means that there is never an optimally
equalized data set (i.e. the same observations for all models and for all neighbourhood sizes). It
also means that comparison of the different neighbourhood results from a single model is ill
advised, in this case, as the observations numbers can be very different, and therefore the model
forecast is being sampled at different locations. Despite this, observation numbers should be
similar when looking at matched spatially sized neighbourhoods from different models if results
are to be compared. One of the main constraints identified through this work is both the sparsity
and geographical distribution of observations throughout the North West Shelf domain, with
several viable locations rejected during the HiRA processing due to their proximity to coastlines.
The purest assessment, in terms of observations, would involve a fixed set of observations,
equalized across both model configurations and all neighbourhoods at every time. This would
remove the variation in observation numbers seen as neighbourhood sizes increase as well as
those seen between the two models and give a clean comparison between two models.
Care should be taken when applying strict equalization rules as this could result in only a small
number of observations being used. The total number of observations used should be large
enough to ensure that the sample is large enough to produce robust results and satisfy rules for
statistical significance. Equalisation rules could also unfairly affect the spatial sampling of the
verification domain. For example, in this study coastal observations would be affected more than
deep ocean observations if neighbourhood equalization were applied, due to the proximity of
the coast.
To a lesser extent, the variation in observation numbers on a day-to-day timescale also has an
impact on any results and could mean that incorrect importance is attributed to certain results,
which are simply due to fluctuations in observation numbers.
The fact that the errors can be reduced through the use of neighbourhoods shows that the ocean
and the atmosphere have similarities in the way the forecasts behave as a function of resolution.
This study did not consider the concept of skill, which incorporates the performance of the
forecast relative to a pre-defined benchmark. For the ocean the choice of reference needs to be
considered. This could be the subject of further work.
To our knowledge, this work is the first attempt to use neighbourhood techniques to assess ocean
models. The promising results showing reductions in errors of the finer resolution configuration
warrant further work. We see a number of directions the current study could be extended.
The study was conducted on daily output which should be appropriate to address eddy mesoscale
variability, but observations are distributed at hourly resolution, and so the next logical step
would be to assess the hourly forecasts against the hourly observation and see how this impacted
the results. This will increase the sample size, if all hourly observations were considered together.
However, it is impossible to speculate on whether considering hourly forecasts would lead to
more noisy statistics, counteracting the larger sample size.
This assessment only looked at SST for this initial examination. Consideration of other ocean
variables would also be of interest, including looking at derived diagnostics such as mixed layer
depth, but the sparsity of observations available for some variables may limit the case studies
available. HiRA as a framework is not remaining static. Enhancements to introduce non-regular
flow-dependent neighbourhoods are planned and may be of benefit to ocean applications in the
future. Finally, an advantage of using the HiRA framework is that results obtained from
deterministic ocean models could also be compared against results from ensemble models when
these become available for ocean applications.

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

## 9. Author contributions
All authors contributed to the introduction, data and methods, and conclusions. RC, JM and MM
contributed to the scientific evaluation and analysis of the results. RC and JM designed and ran
the model assessments. CP supported the assessments through the provision and reformatting
of the data used. MT provided detail on the model configurations used.

## 10.   Competing interests

The authors declare that they have no conflict of interest.

## 11.   Acknowledgements

This study has been conducted using E.U. Copernicus Marine Service Information.
This work has been carried out as part of the Copernicus Marine Environment Monitoring Service
(CMEMS) HiVE project. CMEMS is implemented by Mercator Ocean International in the
framework of a delegation agreement with the European Union.
Model Evaluation Tools (MET) was developed at the National Center for Atmospheric Research
(NCAR) through grants from the National Science Foundation (NSF), the National Oceanic and
Atmospheric Administration (NOAA), The United States Air Force (USAF), and the United States
Department of Energy (DOE). NCAR is sponsored by the United States National Science
Foundation.