# Peer review of "An approach to the verification of high resolution ocean models using spatial methods"

_Ocean Science, 2020_

## Referee Comment (RC1) · Anonymous Referee #1 · 20 Mar 2020

The paper untitled "An approach to the verification of high-resolution ocean models using spatial methods" described a really interesting method to quantify benefit from high resolution model. The paper describes in detail methodology and apply it to compare two ocean circulation forecast models on the Nordic Sea. Scientific results obtain comparing the two forecast system are poorly commented and explained but this scientific analysis is not the main topic of the paper which is really dedicated to the description, implementation of this methodology that was not already applied for ocean forecast. That could be frustrating for readers, authors can certainly add analysis of some results, some suggestions are provided below. Nevertheless, the paper is clear and objectives are well presented and I recommend the publication of this paper if authors take into account few following remarks and comments.

[Figure]

1. Section 2, Figure 1 : this figure presents the domain and the difference of coastline between the two models. Difference of coastline is an important point discussed also latter in the paper and illustrated on fig 4. To be really interesting, I recommend to highlight the differences between the two SST fields on this figure. A more contrasted color bar, for example, can highlight difference of spatial scale, intensity of SST fronts . . . which are the main reasons to apply the HiRA method in this context.

2. Section 3, line 187. Reference to WMO manual is useful but Authors should explained that this guide refers to Atmosphere and that ocean scales are really different. In this paragraph specificities of ocean should be described as difference of scale depending of the areas, open ocean vs shelf, rossby radius . . . This is briefly discussed later in the section (line 245) but it should appear before in the introduction of the method to justify to use it for ocean application.

3. Section 3, fig 3 and 4. Figure 3 and 4 are useful to understand the method and the neightbourhood concept. But it could be really useful to have, on these figures or with a new figure, a clear description (with an example) of how is computed the probability/density function especially in the coastal cases, how the observations are selected in a neightbourhood, where the coastline is different between the two models and when observations are removed from the statistics. A schematic view of this process should be really useful to understand easily some non-intuitive results as for example why there is less observation in a larger domain.

4. Section 4, line 290. I suggest to use zonal and meridional instead of horizontal and vertical.

5. Section 4, figure 4. Unclear or a mistake in the legend. Why a) is 7x7 neightbourhood (NB4) and b) NB5? Comparison should be done between similar neightbourhood.

6. Section 5 and 6, fig 5, 7, 9, 10. It's really difficult to identify differences between each line, probably too much lines on the same figures or more important line should be highlighted (in bold or with darker color?) NB1 and NB2 are the more important, it
is really difficult to distinguished them especially on fig7,9,10. Uncertainty, computed for each line, is difficult to associated to the right line. Is it useful to have the "1" line for AMM15, there is no comparison with AMM7? It's also difficult on these figures to have clear relationship between the uncertainty vertical bar and the difference bar. It will be useful to have on the figure or in a table the information where the difference bars are smaller than the uncertainty. This is discussed in the text (paragraph line 420) but it is difficult to verify what is described on the figures.

7. Section 5. Discussion on the different results obtained on-shelf and off-shelf is really interesting, but in the paper it appears as a mix between feasibility and useful methodology to compare several forecasts and a clear difference due to dynamics, physical ocean process and seasonal cycle. I suggest adding more quantitative information concerning the impact of the number of observation to compute robust statistics. The sentence (line 460) explains that the model are better to forecast open ocean, but is there any impact of the number of observation in the statistics? Do you compute statistics with the same number of observations in the two domain (off-shelf and on-shelf)? Fig 12 and 13 seems to exhibit larger uncertainty in the statistic on-shelf in comparison to off-shelf. On fig 12 and 13, it's clear that main differences between the two models appears in summer. That's not really discussed in the paper, is there clear explanation, is it due to physical seasonal processes or mainly due to the number of observations?

8. Section 5, line 479. Conclusion of this paragraph is not clear. What do you really mean by "closer look at the data"?

9. Section 5, line 508. Last sentence concludes on differences at NB2 scale, could you add comment on this conclusion about significance and robustness of this result.

10. Section 5, fig 14. On this figure lack of observations seems to appear end of May and in the text (line 487) authors indicate that missing data are in April.

---

## Referee Comment (RC2) · Anonymous Referee #2 · 1 Apr 2020

In this contribution, the authors conduct a skill assessment of two operational ocean models running in the North West European Shelf with different configurations and spatial resolutions. Since the increased spatial resolution might require ad hoc metrics to properly reflect the model performance and reduce the impact of so-called double-penalty effects (occurring when using point-to-point comparisons with features present in the model but misplaced with respect to the observations), the present work is welcomed. It addresses this interesting and essential topic by intercomparing models' performances in overlapping regions to infer their respective strengths and weaknesses. Equally, the methodology proposed is consistent and the results obtained are relevant, especially in the framework of the Copernicus Marine Service (albeit not explicitly mentioned in the document).

[Figure]

Based on my expertise on ocean models validation, I particularly appreciate the proposed approach (named HiRA) since it might be useful in parent-son inter-comparisons in order to quantify the added-value of downstream services such as very high resolution coastal models (embedded into CMEMS regional ocean forecasting systems) that are currently running in port-approach areas.

I am confident this work can attract the interest of the scientific audience, being cited in future works dealing with similar issues. The style was fluent although some parts (mainly the introduction and the references) could be revised and enhanced. Based on my judgment, I deem the manuscript acceptable upon minor revision. In the following lines I provide some comments, which should hopefully strengthen even more the manuscript.

General comments:

1. Since the main purpose of this work is to showcase the potential of the proposed methodology in operational ocean forecasting, I miss a reference to the Copernicus Marine Environment Monitoring Service -CMEMS- (Le Traon et al., 2019)., although the in situ observations used here were downloaded from CMEMS catalogue. Within this context, there are some valuable and concerted initiatives such as the Product Quality Working Group (PQWG) or the North Atlantic Regional VALidation tool -NARVAL- (Lorente et al., 2019) where physical and biogeochemical model intercomparisons are conducted on a regular basis to deliver outcomes to a broad scientific community.

Le-Traon et al., 2019. "From Observation to Information and Users: The Copernicus Marine Service Perspective". Front. Mar. Sci., 22, https://doi.org/10.3389/fmars.2019.00234.

Lorente et al., 2019. "The NARVAL software toolbox in support of ocean models skill assessment at regional and coastal scales". Computational Science, ICCS 2019.

2. Equally, I also miss a reference to GODAE Coastal Ocean and Shelf Seas Task

Team (COSS-TT), where the Met Office is an active member, involved in a wealth of valuable initiatives in terms of ocean model inter-comparisons. In this context, I think that the state-of-art about previous inter-comparison exercises is not thorough and is poorly cited, despite of the abundant literature reported elsewhere. In this work, there are only 28 references (which is insufficient) and nearly the 50% of them were published in 2010 or earlier, so an update is highly recommended. Below I suggest a number of recent works to build upon:

Aznar et al, 2015. "Strengths and weaknesses of the CMEMS forecasted and reanalyzed solutions for the Iberia-Biscay-Ireland (IBI) waters". Journal of Marine Systems, 159, 1-14.

Mourre et al., 2019. "Assessment of High-Resolution Regional Ocean Prediction Systems Using Multi-Platform Observations: Illustrations in the Western Mediterranean Sea".

Lorente et al., 2019. "Skill assessment of global, regional, and coastal circulation forecast models: evaluating the benefits of dynamical downscaling in IBI (Iberia–Biscay–Ireland) surface waters". Ocean Science, 15, 967-996. Doi: /10.5194/os-15-967-2019.

Mason et al., 2019. "New insight into 3-D mesoscale eddy properties from CMEMS operational models in the western Mediterranean". Ocean Science, 15, 1111–1131.

Hernández et al., 2018. "Measuring performances, skill and accuracy in operational oceanography: New challenges and approaches". In "New Frontiers in Operational Oceanography", Eds. GODAE OceanView, 759-796, doi:10.17125/gov2018.ch29.

Juza et al, 2015. "From basin to sub-basin scale assessment and intercomparison of numerical simulations in the western Mediterranean Sea". Journal of Marine System, 149, 36-49, doi;10.1016/j.jmarsys.2015.04.010.

Hernández et al., 2015. "Recent progress in performance evaluations and near realtime assessment of operational ocean products". Journal of Operational Oceanogra-

phy, 8, Issue sup2: GODAE OceanView Part 2

Rockel et al., 2015. "The regional downscalling approach: a brief history and recent advances". Curr. Clim. Change Rep., 1, 22–29, https://doi.org/10.1007/s40641-014-0001-3.

Katavouta et al, 2016. "Downscalling ocean conditions with application to the Gulf of Maine, Scotian shelf and adjacent deep ocean". Ocean Model., 104, 54–72.

And some other older works:

Crosnier, L., and C. Le Provost. 2007. "Inter-comparing five forecast operational systems in the North Atlantic and Mediterranean basins: The MERSEA-strand1 methodology". Journal of Marine Systems, 65, 354–375.

Greenberg et al, 2007. "Resolution issues in numerical models of oceanic and coastal circulation". Cont. Shelf Res., 27, 1317–1343.

Hernández, 2011. "Performance of Ocean Forecasting Systems–Intercomparison ProjectsÂů. Book: Operational Oceanography in the 21st Century, Chapter 23.

3. In section 1 (Introduction), a preliminary paragraph about why model inter-comparisons are necessary would be convenient. Equally, a brief description of the types of inter-comparisons exercises would be pertinent: i) between two different forecasting systems in the overlapping region to check the consistency of each model solution; ii) between two versions of the same system, in order to evaluate the added-value of the upgraded one before it is transitioned into fully operational status; iii) a parent-son inter-comparison, to evaluate the quality of the downscaling approach adopted; iv) a comparison between both the forecast and the reanalyzed solutions of the same model in order to infer the primary role of both the grid resolution and the atmospheric forcing, especially in coastal areas (see Aznar et al., 2015, for further details).

4. In section 2.1 (Data and Methods: Forecast), I strongly suggest adding a table to provide a general overview of the two model's main features in a more synthetized way:

version of model, geographic domain, grid resolution, number of depth levels, number of forecast horizons, open boundary conditions, tidal forcing, atmospheric forcing, river forcing, assimilation scheme, bathymetry, etc. Although most of this information is already provided in the text, I think a table would be rather useful as a summary.

5. In section 2.1 (Data and Methods: Forecast), neither river forcing is mentioned, nor river freshwater discharge is taken into account when describing the general considerations. The study-area comprises several rivers estuaries (Seine, Rhine, even Loire) with significant freshwater runoff that might eventually impact on the SST field in coastal areas. Figure 2 shows that some stations are located quite close to those rivers mouth. Please clarify this point, why the river forcing is out of the discussion. In particular, Graham et al (2018) suggested that AMM7 configuration might be more diffusive than AMM15 within river plumes, allowing freshwater input from the Rhine to be advected offshore.

6. In the same line, an event-oriented inter-comparison (with a focus on river plumes and abrupt SST drops due to impulsive-type riverine discharges) would allow you to better infer the ability of each system to capture small-scale coastal processes (with and without HiRA approach). This process-based validation approach, albeit commonly used in meteorology and weather forecasting, is rather novel in operational oceanography and mostly devoted to extreme sea level and wave height episodes. I am not asking to provide new and complementary analysis but please take it as a kind suggestion for future works.

7. With regards to the double-penalty effect, I was somehow expecting a multi-parameter analysis, with a special focus on altimetry products, sea level anomalies and mesoscale eddies. Did you have the chance to test HiRA approach with other variables? If so, could you add a comment about it, even if you only obtained preliminary results? If not, I think this task should remain as a priority for future works and thus be explicitly mentioned in the text.

8. Likewise, I miss a deeper discussion respect to the previous works by Tonani et al (2019) and especially that one by Graham et al (2018) where a "traditional point-to-point SST validation approach" was performed with the new AMM15 system. I think that the fact of contrasting results from both papers / both methodologies could benefit the discussion section, particularly when dealing with on-shelf and off-shelf differences as far as Graham et al (2018) proved the reduction in seasonal SST bias was greater off-shelf than on-shelf when using AMM15 (which supports the results exposed in Figures 9 and 10 of the present work). Again, on-shelf results were worse and you succinctly listed river mixing as a potential source of uncertainties, but no additional information was provided about the role of river forcing (as I aforementioned in point 5). I guess that the river fluxes could have been altered between the two models (being one configuration fresher and cooler than the other).

Minor comments:

Abstract:

I recommend explaining briefly (in two lines) the double penalty effect as part of the potential audience might not be familiarized with this concept. For instance: "[. . .] referred to as the double-penalty effect, occurring in point-to-point comparisons with features present in the model but misplaced with respect to the observations."

Keywords:

I suggest adding "skill assessment", "validation" and/or "double-penalty".

Figure 1:

As previously indicated by the anonymous reviewer 1, a more contrasted color bar is required to highlight the spatial SST differences. Bathymetric contours would be also welcomed.

Figure 8:

Albeit rather obvious, please indicate that masked regions are in grey color.

Introduction:

Lines 58-60: That sentence sounds odd. Could you rephase it, please?

Line 61: please replace "suggested" by "suggesting"

Line 65: please replace "more like" by "more similar to"

Section 2.1. Forecast

Lines 106-108: I guess that hourly instantaneous values are provided for the sea surface and daily averages for the rest of the water column. Please, could you clarify it?

Line 117: Why the study period comprises from January to September 2019? Any chance to expand the analysis to cover the entire 2019 year? That would be interesting to infer seasonal differences between both model configurations. . .

Lines 132-133: please comment that semi-diurnal M2 is one of the predominant tidal constituents in this region (that is the reason to compute means over 25 hours in order to remove the tidal signal).

Section 7: Discussion and conclusions.

Lines 538-539: as previously indicated, provide further insight into on-shelf and off-shelf differences, contrasting the results obtained with those reported in Graham et al (2018).

Lines 540-545: is there any adopted rule or any agreed proposal to wisely select the neighborhood sizes?

---

## Author Comment (AC1) · 22 May 2020

RC – Referee Comment AC – Author Comment MC – Manuscript change

RC - The paper untitled "An approach to the verification of high-resolution ocean models using spatial methods" described a really interesting method to quantify benefit from high resolution model. The paper describes in detail methodology and apply it to compare two ocean circulation forecast models on the Nordic Sea. Scientific results obtain comparing the two forecast system are poorly commented and explained but this scientific analysis is not the main topic of the paper, which is really dedicated to the description, implementation of this methodology that was not already applied for ocean forecast. That could be frustrating for readers, authors can certainly add analy-

sis of some results, some suggestions are provided below. Nevertheless, the paper is clear and objectives are well presented and I recommend the publication of this paper if authors take into account few following remarks and comments.

AC – We thank the reviewer for their time and expertise in reviewing the manuscript. Below are our responses which we hope address the points raised along with changes made to the original manuscript.

RC - 1. Section 2, Figure 1 : this figure presents the domain and the difference of coastline between the two models. Difference of coastline is an important point discussed also latter in the paper and illustrated on fig 4. To be really interesting, I recommend to highlight the differences between the two SST fields on this figure. A more contrasted color bar, for example, can highlight difference of spatial scale, intensity of SST fronts...which are the main reasons to apply the HiRA method in this context.

AC – Agreed. We looked at several different colour palettes which were also colour blind friendly and replotted. In addition, some bathymetry contours were added to address a comment from reviewer 2

MC – New colour scheme used, and bathymetry contours added.

RC - 2. Section 3, line 187. Reference to WMO manual is useful but Authors should explained that this guide refers to Atmosphere and that ocean scales are really different. In this paragraph specificities of ocean should be described as difference of scale de-pending of the areas, open ocean vs shelf, rossby radius...This is briefly discussed later in the section (line 245) but it should appear before in the introduction of the method to justify to use it for ocean application.

AC – Thank you, the WMO reference was indeed only atmospheric, tying in the original justification and application of this method when it was applied to the atmosphere. As such we have expanded the original section to refer to ocean specific characteristics, as well as a brief addition to the introduction.

MC –" A similar principle applies to the ocean, i.e. observations can represent an area around the nominal observation location, though the representative scales are likely to be very different from in the atmosphere. The representative scale for an observation will also depend on local characteristics of the area, for example whether the observation is on the shelf, or in open ocean or likely to be impacted by river discharge."

RC -3. Section 3, fig 3 and 4. Figure 3 and 4 are useful to understand the method and the neighbourhood concept. But it could be really useful to have, on these figures or with a new figure, a clear description (with an example) of how is computed the probability/density function especially in the coastal cases, how the observations are selected in a neighbourhood, where the coastline is different between the two models and when observations are removed from the statistics. A schematic view of this process should be really useful to understand easily some non-intuitive results as for example why there is less observation in a larger domain.

AC – We have added a schematic showing how the neighbourhood points contribute to generating a pdf. We have also expanded the description of how missing points are handled within the text.

MC – added as figure 4

RC - 4. Section 4, line 290. I suggest to use zonal and meridional instead of horizontal and vertical

AC – Accepted

MC – Changed to zonal and meridional

RC - 5. Section 4, figure 4. Unclear or a mistake in the legend. Why a) is 7x7 neighbourhood (NB4) and b) NB5? Comparison should be done between similar neighbourhood.

AC – The idea we were trying to convey was that due to the forecast grids, the kilometre size of neighbourhoods becomes increasingly incorrect as the neighbourhood becomes bigger if simply assuming that multiplying 1.5 km or 7 km are accurate measures of the total size (instead of using the true grid resolution in degrees). Coupled with that is the fact that the model resolution is different in latitudinal and longitudinal directions.

MC – We have separated out and expanded the table describing the neighbourhoods to indicate why a 25x25 AMM15 is more suitable to match to 7x7 AMM7 than the 33x33 AMM15. Also modified the caption to figure 4.

RC -6. Section 5 and 6, fig 5, 7, 9, 10. It's really difficult to identify differences between each line, probably too much lines on the same figures or more important line should be highlighted (in bold or with darker color?) NB1 and NB2 are the more important, is really difficult to distinguished them especially on fig7,9,10. Uncertainty, computed for each line, is difficult to associated to the right line. Is it useful to have the "1" line for AMM15, there is no comparison with AMM7? It's also difficult on these figures to have clear relationship between the uncertainty vertical bar and the difference bar. It will be useful to have on the figure or in a table the information where the difference bars are smaller than the uncertainty. This is discussed in the text (paragraph line 420) but it is difficult to verify what is described on the figures.

AC – Agreed. We felt there was a balance to strike between showing how the scores change with neighbourhood size and the ability to see detail of the actual results. The "1" is important in this case as it shows the default result we would get if HiRA were not being used. However we have tried to make the plots clearer whilst retaining that information.

MC - In order to clarify the plots we have removed some of the larger neighbourhoods from figures 5, 9 and 10. In addition on figure 7 the main lines have been made bold. In order to help with identifying where difference bars are less than the uncertainty, an S has been added over the difference bars where the 95% uncertainly error bars of the

two equivalent lines do not overlap.

RC -7. Section 5. Discussion on the different results obtained on-shelf and off-shelf is really interesting, but in the paper it appears as a mix between feasibility and useful methodology to compare several forecasts and a clear difference due to dynamics, physical ocean process and seasonal cycle. I suggest adding more quantitative information concerning the impact of the number of observation to compute robust statistics. The sentence (line 460) explains that the model are better to forecast open ocean, but is there any impact of the number of observation in the statistics? Do you compute statistics with the same number of observations in the two domain (off-shelf and on-shelf)? Fig 12 and 13 seems to exhibit larger uncertainty in the statistic on-shelf in comparison to off-shelf. On fig 12 and 13, it's clear that main differences between the two models appears in summer. That's not really discussed in the paper, is there clear explanation, is it due to physical seasonal processes or mainly due to the number of observations?

AC – The aim of this paper was to show how the HiRA technique could be used to tease out interesting detail of the model forecasts which could then be a basis for investigation in the future. Notably in this case the apparent seasonal signal. Figure 14 indicates the numbers of observations going into the two domains, and hence the fact that this is a potential source of error. However with the underlying characteristics of the domains being different, it is quite likely that the spatial distribution of observations within the domains is as important as the number of observations. Again, as this was meant to be an investigation of the potential for the verification technique rather than a full model assessment, we did not dig further into the detail of this, but do think it is an important consideration when assessing any results produced using HiRA

RC -8. Section 5, line 479. Conclusion of this paragraph is not clear. What do you really mean by "closer look at the data"?

AC – Essentially breaking the data down and identifying underlying specific parts of the

data which may be contributing to the results counter to the general trend, and which are masked by aggregating.

MC – Edited the text to "therefore a deeper look at the data is required to assess whether this signal is consistent within shorter time periods, or whether there are underlying periods contributing significant and contrasting results to the whole-period aggregate. "

RC -9. Section 5, line 508. Last sentence concludes on differences at NB2 scale, could you add comment on this conclusion about significance and robustness of this result.

AC – Yes, as you indicate the statement is too strong given the error bars presented. We have highlighted that aspect (indicating that the error bars cross, so whilst we cannot say that the difference is significant, we cannot, with the plot provided, say they are not.) And giving suggestions as how to improve this.

MC – "At the NB2 scale, the AMM15 potentially demonstrates more benefit than AMM7 except for April and May, where the two show similar results. There is a balance to be struck in this conclusion as the differences between the two models are rarely greater than the 95% error bars. This in itself does not mean that the results are not significant. However, care should be taken when interpreting such a result as a statistical conclusion rather than broad guidance as to model performance. Attempts to reduce the error bar size, such as increasing the number of observations, or number of times within the period would aid this interpretation."

RC -10. Section 5, fig 14. On this figure lack of observations seems to appear end of May and in the text (line 487) authors indicate that missing data are in April.

AC – The text was incorrect, there was a reduction in observations during May due to issues with the observation extraction from CMEMS. Additionally, there was a forecast reduction in April (due to separate technical issues) not indicated by the plot.

MC – The text is now correct and additionally refers to the missing forecast period.

---

## Author Comment (AC2) · 22 May 2020

RC – Referee Comment AC – Author Comment MC – Manuscript change

RC - In this contribution, the authors conduct a skill assessment of two operational ocean models running in the North West European Shelf with different configurations and spatial resolutions. Since the increased spatial resolution might require ad hoc metrics to properly reflect the model performance and reduce the impact of so-called double-penalty effects (occurring when using point-to-point comparisons with features present in the model but misplaced with respect to the observations), the present work is welcomed. It addresses this interesting and essential topic by intercomparing models' performances in overlapping regions to infer their respective strengths and weak-

nesses. Equally, the methodology proposed is consistent and the results obtained are relevant, especially in the framework of the Copernicus Marine Service (albeit not explicitly mentioned in the document) Based on my expertise on ocean models validation, I particularly appreciate the proposed approach (named HiRA) since it might be useful in parent-son inter-comparisons in order to quantify the added-value of downstream services such as very high resolution coastal models (embedded into CMEMS regional ocean forecasting systems) that are currently running in port-approach areas .I am confident this work can attract the interest of the scientific audience, being cited in future works dealing with similar issues. The style was fluent although some parts (mainly the introduction and the references) could be revised and enhanced. Based on my judgment, I deem the manuscript acceptable upon minor revision. In the following lines I provide some comments, which should hopefully strengthen even more the manuscript.

AR – We thank reviewer 2 for their time and effort reviewing this paper, which produced some very interesting comments. Below we have addressed each point and hope this has resulted in a stronger paper.

RC - General comments: 1. Since the main purpose of this work is to showcase the potential of the proposed methodology in operational ocean forecasting, I miss a reference to the Copernicus Marine Environment Monitoring Service -CMEMS- (Le Traon et al., 2019)., although the in situ observations used here were downloaded from CMEMS catalogue. Within this context, there are some valuable and concerted initiatives such as the Product Quality Working Group (PQWG) or the North Atlantic Regional VALidation tool -NARVAL-(Lorente et al., 2019) where physical and biogeochemical model intercomparisons are conducted on a regular basis to deliver outcomes to a broad scientific community. Le-Traon et al.,2019."From Observation to Information and Users:The Copernicus Marine Service Perspective".Front.Mar.Sci., 22,https://doi.org/10.3389/fmars.2019.00234. Lorente et al., 2019. "The NARVAL software toolbox in support of ocean models skill assessment at regional and coastal

scales". Computational Science, ICCS 2019.

AR -Thankyou, these references have been added where appropriate.

MC – Additional references have been included within the introduction of the paper.

RC - 2. Equally, I also miss a reference to GODAE Coastal Ocean and Shelf Seas Task Team (COSS-TT), where the Met Office is an active member, involved in a wealth of valuable initiatives in terms of ocean model inter-comparisons. In this context, I think that the state-of-art about previous inter-comparison exercises is not thorough and is poorly cited, despite of the abundant literature reported elsewhere. In this work, there are only 28 references (which is insufficient) and nearly the 50% of them were published in 2010 or earlier, so an update is highly recommended. Below I suggest a number of recent works to build upon: Aznar et al, 2015. "Strengths and weaknesses of the CMEMS forecasted and reanalyzed solutions for the Iberia-Biscay-Ireland (IBI) waters". Journal of Marine Systems,159, 1-14. Mourre et al., 2019. "Assessment of High-Resolution Regional Ocean Prediction Systems Using Multi-Platform Observations: Illustrations in the Western Mediterranean Sea". Lorente et al., 2019. "Skill assessment of global, regional, and coastal circulation fore-cast models: evaluating the benefits of dynamical downscaling in IBI (Iberia–Biscay–Ireland) surface waters". Ocean Science, 15, 967-996. Doi: /10.5194/os-15-967-2019. Mason et al., 2019. "New insight into 3-D mesoscale eddy properties from CMEMS operational models in the western Mediterranean". Ocean Science, 15, 1111–1131. Hernández et al., 2018. "Measuring performances, skill and accuracy in operational oceanography: New challenges and approaches". In "New Frontiers in Operational Oceanography", Eds. GODAE OceanView, 759-796, doi:10.17125/gov2018.ch29. Juza et al, 2015. "From basin to sub-basin scale assessment and intercomparison of numerical simulations in the western Mediterranean Sea". Journal of Marine System,149, 36-49, doi;10.1016/j.jmarsys.2015.04.010. Hernández et al., 2015. "Recent progress in performance evaluations and near real-time assessment of operational ocean products". Journal of Operational Oceanography, 8, Issue sup2: GODAE OceanView Part

2 Rockel et al., 2015. "The regional downscalling approach: a brief history and recent advances". Curr. Clim. Change Rep., 1, 22–29, https://doi.org/10.1007/s40641-014-0001-3. Katavouta et al, 2016. "Downscalling ocean conditions with application to the Gulf of Maine, Scotian shelf and adjacent deep ocean". Ocean Model., 104, 54–72. And some other older works: Crosnier, L., and C. Le Provost. 2007. "Inter-comparing five forecast operational systems in the North Atlantic and Mediterranean basins: The MERSEA-strand1 method-ology". Journal of Marine Systems, 65, 354–375. Greenberg et al, 2007. "Resolution issues in numerical models of oceanic and coastalcirculation". Cont. Shelf Res., 27, 1317–1343. Hernández, 2011. "Performance of Ocean Forecasting Systems–Intercomparison ProjectsÂ ÌŁu. Book: Operational Oceanography in the 21st Century, Chapter 23.

AR – Thank you for the additional references. A selection of these have been added to the text in the introduction section to broaden the description of the existing state of things.

MC – Additional references have been included within the introduction of the paper.

RC - 3. In section 1 (Introduction), a preliminary paragraph about why model inter-comparisons are necessary would be convenient. Equally, a brief description of the types of inter-comparisons exercises would be pertinent: i) between two different forecasting systems in the overlapping region to check the consistency of each model solution; ii) between two versions of the same system, in order to evaluate the added-value of the upgraded one before it is transitioned into fully operational status; iii) a parent-son inter-comparison, to evaluate the quality of the downscaling approach adopted; iv) a comparison between both the forecast and the reanalyzed solutions of the same model in order to infer the primary role of both the grid resolution and the atmospheric forcing, especially in coastal areas (see Aznar et al., 2015, for further details).

AR - We have introduced this in combination with some of the references in RC2.

RC - 4. In section 2.1 (Data and Methods: Forecast), I strongly suggest adding a

table to provide a general overview of the two model‒s main features in a more synthetized way: version of model, geographic domain, grid resolution, number of depth levels, number of forecast horizons, open boundary conditions, tidal forcing, atmospheric forcing, river forcing, assimilation scheme, bathymetry, etc. Although most of this information is already provided in the text, I think a table would be rather useful as a summary.

AR - We have added a summary table of the differences relevant for this study.

MC - Additional table (table 1) added in the manuscript

RC- 5. In section 2.1 (Data and Methods: Forecast), neither river forcing is mentioned, nor river freshwater discharge is taken into account when describing the general considerations. The study-area comprises several rivers estuaries (Seine, Rhine, even Loire) with significant freshwater runoff that might eventually impact on the SST field in coastal areas. Figure 2 shows that some stations are located quite close to those rivers mouth. Please clarify this point, why the river forcing is out of the discussion. In particular, Graham et al (2018) suggested that AMM7 configuration might be more diffusive than AMM15 within river plumes, allowing freshwater input from the Rhine to be advected offshore.

AR - Even if it is true that the river forcing plays a role in the coastal areas, it has been proven in Tonani et al. 2019 that it has a very small impact on SST. It is much more evident in surface salinity. We describe in 2.1 the characteristics/differences that are relevant for this study, a comprehensive description of the two forecasting systems is in Tonani et al. 2019. We specify in section 5 that this study is not focused on the coastal areas due to the assumptions in the choice for the neighbour, with different number of observations in the two configurations due mainly to the Land-Sea mask differences. This is a very interesting issue and there is a need for a coastal-focused assessment of the forecast. We will take it account for future work. It is also worth to notice that comparing the model only simulation (non-assimilative analysis) over a long period (30

years) of Graham et al. 2018 with 9 months of assimilative analysis-forecast could be misleading for the reader. Graham et al. experiment is using different lateral boundaries and a different atmospheric forcing and doesn't have data assimilation. Both AMM7 and AMM15 forecasting systems are assimilating SST obs (Insitu and satellite). Even if we consider negligible these differences, it's difficult to justify the comparison of a seasonal mean over 30 years with few months of forecast. The results from Graham et al. 2018 have not been confirmed by Tonani et al. 2019, while assessing the operational trials (with data assimilation and the operational forcing as described in the paper) against OSTIA. This validation is shown at basin level in the paper, but we did analyse also off-shelf and on-shelf differences. There are no significant differences compared to the full domain inter-comparison. From Tonani et al. 2019:

"Temperature RMSD and bias are very small at surface, due to the strong constraint of the data assimilation of SST (as described in 4.3) while at the bottom AMM15 is more accurate in prescribing the temperature at all mooring locations (Table 9). AMM7 and AMM15 both have high salinity errors in the German Bight, as highlighted by the comparison with the buoys that are located closer to the coast (Fino1, Fino3 and UFS-DeBucht). This is most probably due to representation of river discharge. AMM15 performs better than AMM7, probably because it is less diffusive within river plumes and has a lower lateral diffusion. Improved bathymetry and coastal resolution are also likely to play a role in coastal areas with depth less than 20m. AMM15 has halved the salinity error compared to AMM7 when compared with the outer buoys (NsbII and TWEms). It is encouraging to see that AMM15 is better than AMM7 at the bottom at all mooring locations. The decision to use the climatological river discharge dataset instead of E-Hype for AMM7, and subsequently AMM15, has improved salinity remarkably in the German Bight, reducing the model fresh bias. This modification was implemented in April 2017, meaning that we have significantly improved the salinity in the last two major updates of the NWS forecasting system. Nevertheless, using a climatological river runoff dataset is a limitation for a high-resolution forecasting system, affecting variability in coastal water properties. Finding a suitable alternative will be a priority for future

releases of this system."

MC - No changes

RC- 6. In the same line, an event-oriented inter-comparison (with a focus on river plumes and abrupt SST drops due to impulsive-type riverine discharges) would allow you to better infer the ability of each system to capture small-scale coastal processes (with and without HiRA approach). This process-based validation approach, albeit commonly used in meteorology and weather forecasting, is rather novel in operational oceanography and mostly devoted to extreme sea level and wave height episodes. I am not asking to provide new and complementary analysis but please take it as a kind suggestion for future works.

AR - Yes, we agree. We will take this comment into consideration for future work.

MC - no changes

RC- 7. With regards to the double-penalty effect, I was somehow expecting a multi-parameter analysis, with a special focus on altimetry products, sea level anomalies and mesoscale eddies. Did you have the chance to test HiRA approach with other variables? If so, could you add a comment about it, even if you only obtained preliminary results? If not, I think this task should remain as a priority for future works and thus be explicitly mentioned in the text.

AR - Within this assessment we started simple, since we wanted to know whether the technique had anything to offer and only looked at SST, though other parameters were considered (e.g. velocities). One of the next steps will be to apply this to a broader range of parameters. – We have noted this in the conclusions.

MC – The conclusions section has been updated. The conclusion does mention that other variables should be assessed. (lines ∼631) – Agree that more parameters would be good.

RC- 8. Likewise, I miss a deeper discussion respect to the previous works by Tonani et al(2019) and especially that one by Graham et al (2018) where a "traditional point-to-point SST validation approach" was performed with the new AMM15 system. I think that the fact of contrasting results from both papers / both methodologies could benefit the discussion section, particularly when dealing with on-shelf and off-shelf differences as far as Graham et al (2018) proved the reduction in seasonal SST bias was greater off-shelf than on-shelf when using AMM15 (which supports the results exposed in Figures 9 and 10 of the present work). Again, on-shelf results were worse and you succinctly listed river mixing as a potential source of uncertainties, but no additional information was provided about the role of river forcing (as I aforementioned in point5). I guess that the river fluxes could have been altered between the two models (being one configuration fresher and cooler than the other).

AR – Please see also answer to comment 5. The work of this paper is focused on 9 months of forecast validation, while the study discussed in Graham et al 2018 is based on a model only simulation over 30 years. The SST data assimilation has significant impact on both systems (AMM7 and AMM15) due to the good coverage of the observations. The comparison between results of a model only long simulation against few months of forecasts is not straightforward and implies several assumptions that deviates from the object of this paper to assess the forecasts skills in different configurations. We explained in answer 5 that the rivers seem to play a minor role on SST and that we need a specific study focused on the coastal area. The freshwater inflow has for sure an important impact on the stratification and this needs to be properly assessed. The differences on the freshwater are due to horizontal and vertical resolution. Bathymetry and model diffusivity. It is a complex combination of different aspects that is not addressed in this work.

MC – no change

Minor comments:

RC - Abstract: I recommend explaining briefly (in two lines) the double penalty effect as

part of the potential audience might not be familiarized with this concept. For instance: "[...]referred to as the double-penalty effect, occurring in point-to-point comparisons with features present in the model but misplaced with respect to the observations."

AR – Added brief explanation

MC - "…the double-penalty effect. This effect occurs in point-to point comparisons whereby features correctly forecast but misplaced with respect to the observations are penalised twice; once for not occurring at the observed location, and secondly for occurring at the forecast location, where they have not been observed."

RC – Keywords: I suggest adding "skill assessment", "validation" and/or "double-penalty".

AR – Additional keywords will be added

MC – Added 'double penalty' and validation to keywords

RC - Figure 1: As previously indicated by the anonymous reviewer 1, a more contrasted color bar is required to highlight the spatial SST differences. Bathymetric contours would be also welcomed.

AR - Agreed. We looked at a number of different colour palettes which were also colour blind friendly and replotted.

MC – New colour scheme used and bathymetry contours added.

RC - Figure 8: Albeit rather obvious, please indicate that masked regions are in grey color.

AR - Agreed

MC - Added "data within the grey areas is masked" to caption

RC - Introduction: Lines 58-60: That sentence sounds odd. Could you rephase it, please?

[Figure]

AR - Agreed.

MC – Added "In these methods forecasts are assessed at multiple spatial or temporal scales to see how model skill changes as the scale is varied."

RC - Line 61: please replace "suggested" by "suggesting"

AR - Done

MC – "suggested" replaced by "suggesting"

RC - Line 65: please replace "more like" by "more similar to"

AR - Accepted

MC – replaced "more like" with "more similar to"

RC - Section 2.1. Forecast Lines 106-108: I guess that hourly instantaneous values are provided for the sea surface and daily averages for the rest of the water column. Please, could you clarify it?

AR – This is now clarified in the text

MC - "Hourly instantaneous values and daily 25-hour, de-tided, averages are provided for the full water column. "

RC - Line 117: Why the study period comprises from January to September 2019? Any chance to expand the analysis to cover the entire 2019 year? That would be interesting to infer seasonal differences between both model configurations... –

AR – The study period could be expanded to cover a longer period, however since this was an introductory study, we felt that the full benefit of a longer assessment period should also involve additional parameters and a more focussed assessment of the model, rather than, as here, an assessment of the method. The potential seasonal signal gives a focus to any further study.

MC – no change

RC-Lines 132-133: please comment that semi-diurnal M2 is one of the predominant tidal constituents in this region (that is the reason to compute means over 25 hours in order to remove the tidal signal).

AR – The major tidal constituent over the North West European shelf is the semidiurnal lunar component, M2. It has a period of 12 h 25 min (Howarth, M. and Pugh, D.: Chapter 4 Observations of Tides Over the Continental Shelf of North-West Europe, Elsevier Oceanography Series, 35, 135–188, https://doi.org/10.1016/S04229894(08)70502-6, 1983.). The 25 hours mean is therefore removing (or filtering out) the tidal signal.

MC - "The tidal signal is removed because the period of the major tidal constituent, the semidiurnal lunar component M2, is 12hr and 25min (Howarth and Pugh, 1983).

RC - Section 7: Discussion and conclusions. Lines 538-539: as previously indicated, provide further insight into on-shelf and off-shelf differences, contrasting the results obtained with those reported in Graham et al (2018).

AR - See also answers to comment 5 and 8.

MC - Added in the conclusion "Forecast verification studies tailored for the coastal/shelf areas are needed for properly understand the forecast skills in areas with high complexity and fast evolving dynamics."

RC - Lines 540-545: is there any adopted rule or any agreed proposal to wisely select the neighborhood sizes?

AR – There is no accepted rule for doing this, particularly as the appropriate neighbourhood sizes will likely be different for different parameters. As the neighbourhoods become larger the CRPS will (generally) become smaller, but the improvements in the score will occur in smaller increments as the neighbourhood grows. However, there then comes a point where the neighbourhood becomes too large and points are introduced into the neighbourhood for which the observation (which is fixed at a point) is no longer representative, at which point the CRPS tends to increase again (degrade). It is

therefore possible to infer something about the representativeness of the observation and the level of variability within the neighbourhood. In a homogeneous field you could infer aspects of representativeness, but because the sampling is rarely that homogenous (except perhaps deep ocean) it's not easy to infer anything general which can be applied all the time. We investigated whether there were any specific scales that could be identified but could not definitively draw any conclusions. The current method, when applied to the atmosphere, is to initially use a broad set of neighbourhoods and then use a subset of these for routine verification once / if the representativeness for a variable becomes apparent (i.e. the CRPS starts to degrade / tail off).